# Nonlinear Compensation of the Linear Variable Differential Transducer Using an Advanced Snake Optimization Integrated with Tangential Functional Link Artificial Neural Network

**DOI:** 10.3390/s25041074

**Published:** 2025-02-11

**Authors:** Qiuxia Fan, Xinqi Zhang, Zhuang Wen, Lei Xu, Qianqian Zhang

**Affiliations:** School of Automation and Software Engineering, Shanxi University, Taiyuan 030006, China; fanqiuxia0808@sxu.edu.cn (Q.F.); 202222207050@email.sxu.edu.cn (X.Z.); xulei411@sxu.edu.cn (L.X.); zhangqianqian@sxu.edu.cn (Q.Z.)

**Keywords:** functional link artificial neural network, linear variable differential transformer (LVDT), nonlinearity compensation, snake optimization

## Abstract

The linear variable differential transformer is a key component for measuring vibration noise and active vibration isolation. The nonlinear output associated with increased differential displacement in LVDT constrains the measurement range. To extend the measurement range, this paper proposes an advanced Snake Optimization–Tangential Functional Link Artificial Neural Network (ASO-TFLANN) model to extend the linear range of LVDT. First, the Latin hypercube sampling method and the Levy flight method are introduced into the snake optimization (SO) algorithm, which enhances the global search ability and diversity preservation ability of the SO algorithm and effectively solves the common overfitting and local optimal problems in the training process of the gradient descent method. Second, a voltage–displacement test bench is constructed, collecting the input and output data of the LVDT under four different main excitation conditions. Then, the collected input and output data are fed into the ASO-TFLANN model to determine the optimal weight vectors of the tangential functional link Artificial Neural Network (TFLANN). Finally, by comparing with the simulation experiments of several algorithms, it is proven that the ASO proposed in this paper effectively solves the common overfitting and local optimization problems in the training process of the gradient descent method. On this basis, through offline simulation comparison experiments and online tests, it is proven that the method effectively reduces $\epsilon_{fs}$ while expanding the linear range of LVDT and significantly improves the measurement range, which provides a reliable basis for improving measurement range and accuracy.

## 1. Introduction

The linear variable differential transformer (LVDT) is a key component in vibration noise measurement and active vibration isolation. It is extensively employed in domains such as aerospace [1,2], manufacturing [3,4], and industry [5,6] for measuring physical quantities such as distance, position [7], and pressure [8]. The LVDT consists of the primary coil and the two secondary coils that are symmetrically positioned on either side of the primary coil [9]. The primary coil is excited by a 10–50 kHz AC voltage signal and moves linearly between the two oppositely connected secondary coils. The secondary coils generate differential signals based on their relative position to the primary coil. However, as displacement increases, the LVDT exhibits inherent nonlinear input–output characteristics. To guarantee measurement precision, the LVDTs are generally used only within their linear region, which limits their measurement range. Therefore, expanding the measurement range has become a major research focus for experts and scholars worldwide.

Y. Kano et al. [10] proposed a design whose main feature is that the secondary coil is placed obliquely on both sides of the primary coil. Parthasarathi Veeraian et al. [11] introduced a method to extend the linearity range of LVDT by fractional order modelling without significantly affecting sensitivity and linearity. Pipat Prommee et al. [12] utilized logarithmic approximation to perform summation and subtraction operations to eliminate certain LVDT nonlinear characteristics. Hanjari Ram et al. [13] used a dual-slope circuit structure to intelligently process the secondary output under sawtooth wave excitation. Harikumar Ganesan et al. proposed an innovative technique based on an oscillator to control oscillation frequency by adjusting the mutual inductance between the primary and secondary coils [14]. The mutual inductance varies with the motion of the displacement core. The technique requires a microcontroller to generate digital sinusoidal signals and transmit the acquired signals through an analog-to-digital converter. The oscillator approach is further optimized in [15] by adjusting the mutual inductance between the primary and secondary coils in response to the movement of the displacement core to control oscillation frequency. Wandee Petchmaneelumka et al. [16] extended the linear range of LVDT using an inverse function technique. G.Y. Tian et al. [17] presented an equivalent magnetic circuit of LVDT, calculating the mutual inductance, output voltage, and sensitivity. Then, the theory was verified by experimental comparison of two LVDTs with the same structural parameters, as well as with different magnetic materials. These studies focused on mechanical structures or the circuit, enhancing LVDT performance through optimization of mechanical structures and circuit design. However, optimizing mechanical structures and circuits is costly and involves longer design cycles. Unlike fixed mechanical and circuit compensation systems, algorithmic optimization offers greater flexibility, allowing adjustments based on varying operational conditions and application scenarios, such as different frequencies, displacement ranges, or other operating parameters. In contrast, mechanical and circuit systems typically require redesigns or component replacements, making adjustments less flexible.

With the development of neural networks, many scholars have begun to use neural networks to solve this problem. Zhongxun Wang et al. [18] proposed a nonlinear compensation method for LVDT based on a radial-basis function (RBF) neural network. Saroj Kumar Mishra et al. [19] used FLANN to extend the linear range of LVDT and experimentally verified the method’s validity. Based on the previous work, Saroj Kumar Mishra et al. [20] applied the method to two LVDTs and further demonstrated that the method is applicable to any transducer with nonlinear characteristics. Based on the work of Saroj Kumar Mishra [19], Sarita Das et al. proposed a two-stage FLANN network [21]. The paper first uses a low-order FLANN to roughly compensate the nonlinearity of the LVDT model; then a high-order FLANN is used to further compensate the remaining nonlinearity. The results show that the inversion model based on the two-stage FLANN network exhibits higher measurement accuracy and better precision.

However, FLANN uses the gradient descent method to adjust parameters, coming with problems such as slow convergence and reliance on gradient information. To address these limitations, experts have explored various bionic optimization algorithms. However, there is only one example in the field of LVDT output optimization. Li Minghui et al. [22] proposed a BP neural network optimized with the ant colony algorithm to compensate for the nonlinear output of the LVDT. This approach leverages the ant colony algorithm to search for the optimal ranges of neural network weights and thresholds, thereby overcoming common BP neural network shortcomings, such as susceptibility to local minima and slow convergence. The study provides valuable insights for this article, which employs a meta-heuristic algorithm to further improve the compensation process for the nonlinear output of the LVDT. Nevertheless, all of these studies used only a limited number of data points, a dozen, and all of the displaced data points were integers.

Hashim Fatma A. [23] introduced a new metaheuristic approach known as the snake optimization (SO) algorithm. By modelling the unique behaviors of snakes—such as foraging, fighting, mating, and laying eggs—as mathematical processes like global search, diversity maintenance, and the introduction of randomness, the algorithm aims to discover optimal solutions within a defined search space. The SO algorithm does not rely on gradient information and has the advantage of high computational efficiency, which has been applied to the optimization process of the power grid [24]. Ibrahim AlShourbaji et al. [25] applied the SO algorithm to feature selection, which reduces the chances of traditional methods falling into the local optimum. Qilin Li et al. [26] applied the SO algorithm to path planning to reduce the problem of premature convergence of traditional methods. Building upon previous studies in the field of snake optimization [23,25,26], this study proposes a method to refine the weight coefficient matrix of the controller. By leveraging insights from existing research, our work applies the SO algorithm to address the limitations of the linear quadratic regulator (LQR) in vehicle active suspension systems, where defining the weight coefficient matrices Q and R is often subjective and inefficient. Comparative simulations and experiments have demonstrated that the SO algorithm effectively optimizes the LQR controller weight coefficient matrix [27].

This paper adopts SO theory to solve the problems of relying on gradient information and low computational efficiency encountered when FLANN utilizes the gradient descent method to adjust the parameters between input and output layers. However, the optimization effect of SO is easily affected by the initial population. If the initial population is not properly selected, it may cause the search to fall into a local optimum, which in turn affects the global search capability of the algorithm and the accuracy of the final results. Hence, Latin hypercubic sampling (LHS) is employed to initialize the population and enhance the diversity of the snake population. In addition, the selection of the step size of SO depends on the current solution position, which can easily lead to a local optimum. Thus, Levy flight is adapted to generate the step size, resulting in a more randomized step size to avoid falling into local optimization. An adaptive inverse model based on SO and Tangential Functional Link Artificial Neural Network (TFLANN) is proposed to achieve nonlinear compensation of LVDT and to broaden the measurement range of LVDT. The contributions of this paper can be summarized as follows.

Based on the SO algorithm, the population diversity is optimized by LHS using Levy flight generating the step size to reduce the impact of the SO algorithm’s species parameter settings on the optimization results and jump out of the local optimum through a larger step size, which makes it more likely to converge to the globally optimal solution;Introducing tangent function into FLANN to construct the inverse model of LVDT;A large number of comparative simulation experiments are conducted between the ASO algorithm and other algorithms to verify the superiority of ASO in dealing with single-peak function and multi-peak function problems;Offline comparative simulation experiments are conducted between ASO-TFLANN and other methods to verify the effectiveness of ASO-TFLANN in extending the linear range of LVDT; online experiments are conducted using ASO-TFLANN to verify the feasibility of ASO-TFLANN in extending the linear range of LVDT.

The remainder of this paper is organized as follows. Section 2 describes the working principle of LVDT. Section 3 discusses the study of the ASO-TFLANN method for the nonlinear compensation of LVDT. Section 4 compares the performance of the proposed ASO algorithm with seven other algorithms and details the experimental program, including the training of the ASO-TFLANN model, the offline comparative simulation tests using the experimental data, and the subsequent online simulation tests on the trained ASO-TFLANN model. Finally, Section 5 presents the conclusions.

## 2. LVDT Working Principle

Figure 1 illustrates the operating principle of the LVDT. Unlike traditional LVDT sensor structures, which use a movable core to produce displacement, the LVDT is designed with a primary coil ($C_{p}$) and a secondary coil group ($C_{s1}$ and $C_{s2}$). In this configuration, the primary coil ($C_{p}$) is completely enclosed by the secondary coil group, ensuring uniform coupling and improved measurement stability. The primary coil has $n_{p}$ turns and a length of $L_{p}$, where $n_{p}$ refers to the number of turns and $L_{p}$ is the physical length of the coil. The secondary coils ($C_{s1}$ and $C_{s2}$) are symmetrically positioned at the midpoint of $C_{p}$, each with $n_{s}$ turns and a length of $L_{s}$, where $n_{s}$ and $L_{s}$ represent the number of turns and the length of the secondary coils, respectively. All coils are constructed with eight layers of 36AWG (diameter 0.127 mm) polyimide-coated wire, and each coil has a thickness of 2 mm. The primary coil $C_{p}$ is typically excited by a sinusoidal AC signal in the frequency range of 10 to 50 kHz, with an effective voltage of 5–15 V, which induces an alternating current through the secondary coils. In the stationary state, when the primary coil ($C_{p}$) is centered at the midpoint, the induced voltage in the secondary coils is ideally zero due to symmetry. The time-varying magnetic field from the primary coil induces equal and opposite voltages in the two secondary coils, leading to cancellation of the signals. As a result, the secondary coil group does not generate a net sinusoidal voltage signal. However, when there is any displacement between the coils, this symmetry is broken, and the secondary coils generate differential sinusoidal voltage signals that are proportional to the displacement. However, when there is a small relative displacement between the primary and secondary coils, the secondary coil group induces differential sinusoidal voltage signals, where the amplitude is proportional to the differential displacement between $C_{p}$ and the secondary coil group. Nonlinearity occurs when there is a relatively large relative displacement between the primary and secondary coils. Additionally, if the coils move in the opposite direction, the phase of the induced sinusoidal voltage signal shifts by 180°.

According to Faraday’s law of electromagnetic induction, the induced voltages of $C_{s1}$ and $C_{s2}$ are denoted as (1) and (2), respectively: (1)$$\begin{array}{cc} V_{S1} & =k\left(\frac{d\Phi_{S1}}{dt}\right) \end{array}$$(2)$$\begin{array}{cc} V_{S2} & =-k\left(\frac{d\Phi_{S2}}{dt}\right) \end{array}$$
where *k* is a constant of the proportionality. $\Phi_{S1}$ and $\Phi_{S2}$ are the magnetic fluxes of the secondary coils $C_{s1}$ and $C_{s2}$, respectively, as shown in (3) and (4): (3)$$\begin{array}{cc} \Phi_{S1} & =\Phi_{0}\sin\left(\omega t+\theta_{1}\right) \end{array}$$(4)$$\begin{array}{cc} \Phi_{S2} & =\Phi_{0}\sin\left(\omega t+\theta_{2}\right) \end{array}$$
where $\Phi_{0}$ is the maximum magnetic flux. ω is the angular frequency of the AC signal. $\theta_{1}$ is the phase angle of $C_{s1}$, and $\theta_{2}$ is the phase angle of $C_{s2}$.

Using a Taylor expansion for the sinusoidal functions of the magnetic fluxes $\Phi_{S1}$ and $\Phi_{S2}$, (1), (2) is deformed as(5)$$\begin{array}{cc} V_{S1} & =k_{11}x_{d}+k_{12}x_{d}^{2}+k_{13}x_{d}^{3} \end{array}$$(6)$$\begin{array}{cc} V_{S2} & =k_{21}\left(-x_{d}\right)+k_{22}{\left(-x_{d}\right)}^{2}+k_{23}{\left(-x_{d}\right)}^{3} \end{array}$$
where $k_{11}$ and $k_{21}$ are linear proportionality constants; $k_{12}$ and $k_{22}$ are nonlinear proportionality constants for the quadratic term; $k_{13}$ and $k_{23}$ are nonlinear proportionality constants for the cubic term; $x_{d}$ is the differential displacement value. The differential voltage of the two secondary coils $V_{s}$ is represented as follows:(7)$$\begin{array}{cc} V_{s} & =V_{S1}-V_{S2}\\ \; & =k_{11}x_{d}+k_{12}x_{d}^{2}+k_{13}x_{d}^{3}-\left[k_{21}\left(-x_{d}\right)+k_{22}{\left(-x_{d}\right)}^{2}+k_{23}{\left(-x_{d}\right)}^{3}\right]\\ \; & =\left(k_{11}+k_{21}\right)x_{d}+\left(k_{12}-k_{22}\right)x_{d}^{2}+\left(k_{13}+k_{23}\right)x_{d}^{3} \end{array}$$ Differential voltage $V_{s}$ is linearly proportional to displacement $x_{d}$. It is effective for measuring small displacements within the linear range of the LVDT. However, as displacement $x_{d}$ increases, the higher-order terms become more important, leading to non-linear output voltage. Therefore, non-linear compensation methods are necessary to extend the accurate measurement range of LVDT.

## 3. ASO-TFLANN Method for Nonlinear Compensation of LVDT

### 3.1. Snake Optimizer

For the snake optimization algorithm, the snake’s foraging and mating behaviors are mainly affected by the amount of food and the ambient temperature. Based on the conditions of insufficient and sufficient food, the algorithm is divided into two phases of global and local search, and the amount of food $Q_{u}\left(t\right)$ and the ambient temperature $T_{p}\left(t\right)$ are defined as (8) and (9), respectively: (8)$$\begin{array}{cc} Q_{u}\left(t\right) & =0.5e^{\frac{t-T}{T}} \end{array}$$(9)$$\begin{array}{cc} T_{p}\left(t\right) & =e^{\frac{-t}{T}} \end{array}$$
where *t* signifies the current iteration count and *T* indicates the maximum number of iterations.

#### 3.1.1. Global Search Phase (Food Scarcity)

When $Q_{u}\left(t\right)\leq Q_{ulim}$ ($Q_{ulim}$ is the food threshold), the algorithm enters the global exploration phase, where both male and female individuals choose random positions to search for food, and the positions are optimized as (Equation 10) and (Equation 11):(10)$$\begin{array}{cc} Xf_{i,j}\left(t\right) & =Xf_{rd,j}\left(t-1\right)\pm 0.05e^{\left(-Fit\left(Xm_{rd}\right)\right)/Fit\left(Xm_{i}\right)}\left(\left(X_{i,j}^{max}-X_{i,j}^{min}\right)rd+X_{i,j}^{min}\right) \end{array}$$(11)$$\begin{array}{cc} Xm_{i,j}\left(t\right) & =Xm_{rd,j}\left(t-1\right)\pm 0.05e^{\left(-Fit(Xf_{rd})\right)/Fit\left(Xf_{i}\right)}\left(\left(X_{i,j}^{max}-X_{i,j}^{min}\right)rd+X_{i,j}^{min}\right) \end{array}$$
where $\left(X_{f_{i,j}}\left(t\right)\right)\left(X_{m_{i,j}}\left(t\right)\right)$ denotes the value at dimension *j* for the *i*th female(male) snake individual in generation *t*. $X_{f_{rd,j}}\left(t-1\right)$ ($X_{m_{rd,j}}\left(t-1\right)$) indicates the position value at dimension *j* for a randomly selected individual guiding the female (male) population in generation t−1. $X_{m_{rd}}$($X_{f_{rd}}$) refers to the position value of a randomly selected individual guiding the male (female) population, with a corresponding fitness $Fit(X_{m_{rd}})$($Fit(X_{f_{rd}})$). $X_{m_{i}}$ represents the *i*th male snake individual, with its fitness $Fit(X_{m_{i}})$, and $X_{f_{i}}$ denotes the *i*th female individual, with its fitness $Fit(X_{f_{i}})$. Additionally, rd is a number drawn from a uniform distribution within the range [0,1]. $X_{i,j}^{max}$ and $X_{i,j}^{min}$ denote the upper and lower bounds of dimension *j* for the *i*th individual, respectively.

#### 3.1.2. Local Search Phase (Adequate Food)

When $Q_{u}\left(t\right)>Q_{ulim}$ and $T_{p}\left(t\right)>T_{plim}$ ($T_{plim}$ is the temperature threshold), the snake population transitions into a foraging state, signaling the algorithm to enter a local search phase. During this phase, both male and female individuals perform a search around the position of the global best individual, $X_{fd}$. This process is illustrated in (12) and (13): (12)$$\begin{array}{cc} X_{f_{i}}\left(t+1\right) & =X_{fd}\left(t\right)\pm 2rd\;T_{p}\left(t\right)\left(X_{fd}\left(t\right)-X_{f_{i}}\left(t\right)\right) \end{array}$$(13)$$\begin{array}{cc} X_{m_{i}}\left(t+1\right) & =X_{fd}\left(t\right)\pm 2rd\;T_{p}\left(t\right)\left(X_{fd}\left(t\right)-X_{m_{i}}\left(t\right)\right) \end{array}$$
where $X_{fd}\left(t\right)$ represents the position of the best individual in generation *t* of the snake population, $Xm_{i}\left(t\right)$$\left(Xf_{i}\left(t\right)\right)$ represents the position of the *i*th male(female) snake in generation *t* of the snake population.

When $Q_{u}\left(t\right)>Q_{ulim}$ and $T_{p}\left(t\right)\leq T_{plim}$, the snake population enters a fighting or mating state. During the fighting state, the position update formula for male and female individuals is as in (14) and (15): (14)$$\begin{array}{cc} X_{m_{i}}\left(t+1\right) & =X_{m_{i}}\left(t\right)+2e^{-\frac{Fit(X_{f_{bt}})}{Fit(X_{m_{i}})}}rd\left(Q_{u}X_{f_{bt}}\left(t\right)-X_{m_{i}}\left(t\right)\right) \end{array}$$(15)$$\begin{array}{cc} X_{f_{i}}\left(t+1\right) & =X_{f_{i}}\left(t\right)+2e^{-\frac{Fit(X_{m_{bt}})}{Fit(X_{f_{i}})}}rd\left(Q_{u}X_{m_{bt}}\left(t\right)-X_{f_{i}}\left(t\right)\right) \end{array}$$
where $Xm_{bt}\left(t\right)$ and $Xf_{bt}\left(t\right)$, respectively, represent the position of the best individual in generation *t* of the male and female snake populations.

When in the mating state, the position update for male and female individuals is shown in (16) and (17): (16)$$\begin{array}{cc} X_{f_{i}}\left(t+1\right) & =X_{f_{i}}\left(t\right)+2M_{f_{i}}rd\left(Q_{u}X_{m_{i}}\left(t\right)-X_{f_{i}}\left(t\right)\right) \end{array}$$(17)$$\begin{array}{cc} X_{m_{i}}\left(t+1\right) & =X_{m_{i}}\left(t\right)+2M_{m_{i}}rd\left(Q_{u}X_{f_{i}}\left(t\right)-X_{m_{i}}\left(t\right)\right) \end{array}$$
where $M_{f_{i}}$ and $M_{m_{i}}$, respectively, represent the mating capabilities of the *i*th female and male individuals. If snake eggs hatch, the selection of the weakest male and female for elimination is replaced by Equations (18) and (19): (18)$$\begin{array}{cc} X_{w,m} & =X_{min}+rd\left(X_{max}-X_{min}\right) \end{array}$$(19)$$\begin{array}{cc} X_{w,f} & =X_{min}+rd\left(X_{max}-X_{min}\right) \end{array}$$
where $X_{w,m}$ ($X_{w,f}$) is the worst male (female) individual. $X_{min}$ is the lower bound of feasible solution *X*, and $X_{max}$ is the upper bound of feasible solution *X*.

### 3.2. Advanced Snake Optimization Algorithm

SO uses a stochastic approach to initialize the snake population, which is a pseudo-random initialization, leading to an uneven distribution of the snake population in the solution space. To overcome this problem, in this article, the Latin hypercube sampling (LHS) method is used in the snake population initialization process to make the distribution of the snake population more uniform. Unlike simple random sampling, LHS ensures that the values of the variables in each dimension cover the entire search space uniformly, thus improving the sampling efficiency and representativeness. When sampling *n* points in a *d*-dimensional space is recorded, each dimension of the sampling space is defined as the interval [0, 1] and wish to generate $X\in R^{n\times d}$, where each row of *X* denotes a sampled point, and each dimension covers [0, 1] uniformly. This method improves the global search by segmenting the values within each dimension to ensure that the samples cover the entire search space. The basic procedure of the Latin hypercube sampling method is as follows:Each scene dimension is divided into n equal partitions according to the cumulative density function (*n* is the scene element dimension).Data points are randomly selected within a single-dimensional partition.The results of each dimension are merged to generate the sampling space.The desired samples in the sampling space are selected using a random method.

Figure 2a,b show the Latin hypercubic sampling distribution and random distribution, respectively. The LHS ensures that each dimension is uniformly covered and that the data points show a more even distribution along the *X* and *Y* axes, with relatively uniform distances between points, avoiding any obvious clustering or sparsity. This method improves the ability of global search by dividing each dimension into multiple intervals, thus ensuring that the samples can fully cover the search space.

In the SO, the step size of the snake’s movement is usually determined by the current position of the individual and the global optimal position. The step size is relatively small and tends to be continuous, and the snake moves more smoothly in the solution space, which makes it easy to fall into the local optimum. To solve this problem, Levy flight is introduced as a stochastic search method conforming to the Levy distribution. The hybrid search behavior of the Levy flight combines short-range and occasionally long-range searches, which endows the algorithm with a powerful global search capability. In snake optimization algorithms, using random step sizes generated by Levy flights ensures a certain level of convergence accuracy. It also provides the opportunity to jump out of the local optimum through larger step sizes, thus making it more likely to converge to a globally optimal solution. The steps of introducing Levy flight into the snake optimization algorithm are similar to the original algorithm, except that at each iteration, the step size *s* is no longer a random value generated by the system but is generated by the Levy flight law. The snake’s position update formula thus become (20) and (21): (20)$$\begin{array}{cc} X_{f_{i,j}}\left(t+1\right) & =X_{f_{rd,j}}\left(t\right)\pm 0.05e^{\left(-\frac{Fit(X_{m_{rd}})}{Fit(X_{m_{i}})}\right)}s \end{array}$$(21)$$\begin{array}{cc} X_{m_{i,j}}\left(t+1\right) & =X_{m_{rd,j}}\left(t\right)\pm 0.05e^{\left(-\frac{Fit(X_{f_{rd}})}{Fit(X_{f_{i}})}\right)}s \end{array}$$
where *s* is the Levy flight step, given by (22). $u∼N(0,\sigma^{2})$, v∼N(0,1), and β is taken to be 1.5. The variance σ is given by (23)(22)$$\begin{array}{cc} s & =\frac{u}{{\left|v\right|}^{1/\beta }} \end{array}$$(23)$$\begin{array}{cc} \sigma & ={\left(\frac{\Gamma \left(1+\beta \right)\sin\left(\frac{\pi \beta }{2}\right)}{\beta \Gamma \left(\frac{1+\beta }{2}\right)2^{\frac{1-\beta }{2}}}\right)}^{\frac{1}{\beta }} \end{array}$$ Using the Advanced Snake Optimization Algorithm (ASO) to update $W^{T}$, Algorithm 1 shows the pseudo-code of the ASO algorithm to optimize $W^{T}$.

**Algorithm 1** Pseudo-code of the ASO algorithm to optimize *W^T^*.1Initialize the problem setting.2Randomly initialize male and female populations.3Define Temp using Equation (9).4**While** ($T_{p}\left(t\right)<T_{plim}$) **do**5    Evaluate each group $X_{m}$ and $X_{f}$.6    Find the best male $X_{fbt}$ and $X_{mbt}$.7    Define food quantity $Q_{u}\left(t\right)$ using Equation (Equation 8).8    **If** ($Q_{u}\left(t\right)<Q_{umin}$)9        Perform exploration using Equations (20) and (21).10    **Else if** ($Q_{u}\left(t\right)>Q_{umin}$)11        Perform exploitation using Equations (12) and (13).12    **Else**13        **If** (rand >0.6)14           Snakes in fight mode using Equations (14) and (15).15        **Else**16           Snakes in mating mode using Equations (16) and (17).17           Change the worst male and female using Equations (18) and (19).18        **End if**19    **End if**20
**End while**
21Return the best solution.

### 3.3. TFLANN Model Construction of LVDT

In this subsection, the displacement-voltage data of the LVDT are obtained through experiments and expanded with higher-order tangent to obtain the nonlinear mapping values. Based on the ASO and the TFLANN, an adaptive inverse model of the LVDT is established. The weights between the input and output layers of the TFLANN by employing the methods of the snake optimization theory, such as global searching, diversity maintenance, and stochastic introduction, etc., are optimized to solve the overfitting and local optimization problems that occur when using the gradient descent method. The solution for obtaining the optimal weight vector of TFLANN to realize the LVDT nonlinear compensation and its working principle are shown in Figure 3.

The functional link processes either an individual element of a pattern or the entire pattern by creating a set of linearly independent functions, which are then assessed using the pattern as the input argument [28,29]. To prevent the tangent function from exceeding the definition domain, $V_{s}$ is scaled by coordinates to obtain $V_{r}$, which is inputted into TFLANN to perform a fifty-dimensional tangent function chain expansion to obtain the nonlinear mapping values, which are used together with $V_{r}$ as inputs to the model, and the function chain expansion is shown in (Equation 24):(24)$$\begin{array}{c} s_{i}=\left\{\begin{array}{cc} V_{r}, & \mathrm{if}\;i=1\\ \frac{\tan(V_{r})}{i-1}, & \mathrm{if}\;1<i<P \end{array}\right. \end{array}$$
where *P* denotes the dimensionality of the neural network’s input layer. $s_{i}$ represents the *i*th neuron;

Building a combined prediction model with optimized weights:(25)$$\begin{array}{cc} y & =\sum_{i=1}^{Q}w_{i}s_{i}=SW^{T} \end{array}$$
where *y* denotes the final prediction result. $w_{i}$ is the weight occupied by a single neuron. *S* is the prediction model input layer vector. $W^{T}$ is the transpose of the optimal weight vector of the prediction model input layer.

In addition, it converts virtual displacements accepted by the LVDT voltage–displacement test bench into actual displacement values:(26)$$\begin{array}{cc} X_{S} & =\frac{X_{c}-X_{z}}{R} \end{array}$$
where $X_{s}$ denotes the actual displacement. $X_{z}$ signifies the grating sensor reading when the differential voltage output of the secondary coil group of the LVDT is zero. *R* denotes the resolution of the grating sensor’s encoder.

The quality of individual positions in the snake population is evaluated by the fitness function $Fit\left(X_{i}\right)$. For ease of handling, the fitness function is chosen as (Equation 27)(27)$$\begin{array}{cc} Fit(X_{i}) & =\frac{1}{2n}\sum_{i=1}^{n}{\left(y_{a}\left(i\right)-y\left(i\right)\right)}^{2} \end{array}$$
where $y_{a}\left(i\right)$ represents the actual displacement measured by the LVDT during the *i*th experimental trial. y(i) is the predicted output derived from the same input. *n* denotes the total number of distinct displacement experiments conducted.

## 4. Numerical Simulation and Experimental Verification

The local and global optimization capabilities of the ASO and SO algorithms on single-peak and multi-peak functions are evaluated through 48 independent comparison experiments between other algorithms under the same conditions using six benchmark functions. To verify the advantages of the ASO-TFLANN nonlinear compensation method, the LVDT voltage–displacement test bench was constructed for testing, and the displacement–voltage data of the LVDT under four different frequencies of primary excitation were obtained. The displacement and voltage data of the LVDT under the primary excitation of the four different frequencies were fed to the four models for iterative training, which are the SO-TFLANN, the Advanced Snake Optimization–Sine–Cosine Functional Linked Network Chain (ASO-SCFLANN), the Tangent Functional Linked Artificial Neural Network (TFLANN), and the Sine–Cosine Functional Artificial Neural Linked Network (SCFLANN). The optimal parameters obtained from the training are given to the above four models for comparative analysis in simulation experiments.

### 4.1. ASO Performance Test

To evaluate the optimization performance of ASO, Particle Swarm Optimization (PSO), Ant Lion Optimization (ALO), and Moth Flame Optimization (MFO) by six benchmark functions with a swarm size of 30 and an iteration number of 500, Sine–Cosine Algorithm (SCA), Salp Swarm Algorithm (SS), Chameleon Search Algorithm (CSA), snake optimization (SO), and ASO conducted 70 independent comparison experiments. The six benchmark functions are listed in Table 1. F1, F2, and F3 functions are single-peak functions that can be used to test the local optimization ability of the algorithms. F4, F5, and F6 functions are multi-peak functions that can be used to test the global optimization ability of the algorithms. The resultant data use the mean value which measures the accuracy of the algorithm. The above algorithm was run on Lenovo Savior R7000P with the Windows 10 system and MATLAB version 2023b.

The test data of the eight algorithms for the above benchmark functions are given in Table 2. Figure 4, Figure 5, Figure 6, Figure 7, Figure 8, Figure 9, Figure 10, Figure 11, Figure 12, Figure 13, Figure 14 and Figure 15 display 3D plots of the benchmark functions F1, F2, F3, F4, F5, and F6, along with the average fitness curves of the algorithms following optimization. Figure 5 shows that ASO, SO, and SS optimize the F1 function better compared to PSO, ALO, MFO, SCA, and CSA. At about the 250th iteration, the value of the objective function for ASO drops dramatically, far below that of the other algorithms, and eventually reaches a value close to the order of magnitude of −100. Among them, ASO is 1000 times more accurate than SO and 90 orders of magnitude higher than SS. As shown in Figure 7, ASO shows outstanding performance throughout. Initially, all algorithms have relatively close objective function values, but as the number of iterations increases, ASO quickly shows a significant advantage after about 300 iterations, with the objective function value rapidly decreasing to be much lower than the other algorithms. At the end of 500 iterations, the objective function value of ASO is close to $10^{-45}$, and the objective function value of SO is close to $10^{-43}$, indicating that ASO has very high accuracy and excellent convergence on this function. In contrast, the other algorithms perform mediocrely on the F2 function, especially the objective function values of MFO and CSA decrease only slightly in the late iteration. Taken together, ASO shows excellent global search capability on the F2 test function and can quickly converge to very small values, far exceeding the performance of the other algorithms. In the optimization tests for the F3 function, ASO still had the best results, followed by SO (see Figure 9). At the end of 250 iterations, SO drops rapidly, and the value is the same as that of ASO; at the end of 400 iterations, ASO drops rapidly again to 2.6087, breaking through the local optimum. Comprehensively, ASO shows excellent global search ability in the late iteration of the F3 test function using Levy flight, which can break through the local optimum and far outperforms the performance of other algorithms. Figure 11 shows that in the F4 function optimization test, both ASO and SO reach the theoretical optimal value of 0 on average, but ASO finds the optimal solution relatively faster; the second-ranked PSO has an average of 11, and SS and MFO are both in the third place with 14. Figure 13 shows that ASO, SO, and CSA outperform PSO, ALO, MFO, SCA, and SS in the optimization test for F5 functions. ASO has the value of the objective function that is three orders of magnitude lower than that of SO, after four fast drops at iterations 250-350. At iterations 450-500, there is one fast drop for both SO and ASO, but ASO is still 100 times more accurate than SO. Figure 15 shows that ASO, SO, and CSA are better optimized in the optimization test of the F6 function, and ASO and SO have the same performance. Overall, the convergence trend of ASO is more obvious from Figure 4 to Figure 15. Especially when the number of iterations exceeds 200, the average fitting curves of F3 and F5 of ASO can find better solutions than the other algorithms after several step-downs. This indicates that ASO has faster convergence speed and higher search accuracy than SO and other algorithms, and is less likely to fall into local optimization, thus showing stronger competitiveness.

### 4.2. ASO-TFLANN Model Training and Comparative Simulation

The LVDT voltage–displacement test bench consists of a high-precision stepping motor, a grating sensor, a stepping motor driver, a signal generator, an oscilloscope, and DC power supply. High-precision stepping motor’s stepping angle is 1.8°/step, the lead range is 1 mm, repeatable positioning accuracy is ±0.005 mm. The stepping motor is connected with the primary coil of LVDT. The grating sensor’s pitch is 20 µm, accuracy is 1 µm, the displacement signal is fed back to the stepping motor to achieve closed-loop regulation, model ATOM4TO-300. The stepper motor driver converts the displacement command into a stepping angle, enabling the precise control of the stepper motor in a closed-loop system. The signal generator (model DG1022G) provides sinusoidal voltage signals to the primary coil of the LVDT. The computer sends non-integer displacement commands to the stepper motor driver to generate the desired displacement of the LVDT’s primary coil, which helps avoid integer displacement values in the test results. The oscilloscope (model DS1102E) is used to record the differential voltage across the secondary coil, capturing the secondary displacement of the stepper motor. A visual representation of the setup used for the tests at four different frequencies, namely 10 kHz, 20 kHz, 30 kHz, and 50 kHz, is shown in Figure 16. The same setup was used for the subsequent online experiments as well.

Under the primary excitation of four different frequencies, namely 10 kHz, 20 kHz, 30 kHz, and 50 kHz, 17 sets of experimental data for each frequency are presented in Table 3. $X_{c}$ is the virtual displacement accepted by the LVDT voltage–displacement test bench, and is dimensionless; $V_{s}$ is the secondary coil differential voltage in volts. The displacement-voltage output curve is shown in Figure 17.

To verify the advantages of the ASO-TFLANN nonlinear compensation method, the collected voltage–displacement data of four primary excitations at 10 kHz, 20 kHz, 30 kHz, and 50 kHz were input into four models (such as ASO-TFLANN, SO-SCFLANN, TFLANN, and SCFLANN) for offline comparative simulation experiments and analysis. The error ϵ, maximum absolute error $\epsilon_{max}$, and maximum full-scale error $\epsilon_{fs}$ [30] of the four methods in the LVDT measurement range were obtained through Equations (Equation 28)–(Equation 30). Table 4 shows the $\epsilon_{max}$ of the four methods at different frequencies, and Table 5 shows the $\epsilon_{fs}$ of the four methods at different frequencies.(28)$$\begin{array}{c} \epsilon =X_{s}-y \end{array}$$(29)$$\begin{array}{c} \epsilon_{\max}=\max\left(\epsilon \right) \end{array}$$(30)$$\begin{array}{cc} \epsilon_{fs} & =\frac{\epsilon_{max}}{\mathrm{stroke}\;\mathrm{range}}\times 100\% \end{array}$$

Figure 18a shows the comparison of output error images of different methods at 10 kHz. From Figure 18a, Table 4 and Table 5, it can be seen that when the primary excitation is 10 kHz, the $\epsilon_{max}$ using ASO-TFLANN, ASO-SCFLANN, and TFLANN are 84.40 µm, 96.89 µm, 365.74 µm, respectively, which are lower than the 560.13 µm using SCFLANN, and the ASO-TFLANN calculates the smallest $\epsilon_{fs}$ of 0.61%, which is 84.9% lower than that of SCFLANN, followed by ASO-SCFLANN and TFLANN, which are 82.7% and 34.7% lower, respectively.

Figure 18b shows the comparison of output error images of different methods at 20 kHz. As shown in Figure 18b, Table 4 and Table 5, when the primary excitation is 20 kHz, the SCFLANN’s $\epsilon_{max}$ is 668.33 µm, the ASO-TFLANN’s is the smallest, 42.41 µm, which is 93.65% lower than that of the SCFLANN, the ASO- SCFLANN is the next smallest at 150.62 µm, which is 77.46% lower, and the TFLANN’s $\epsilon_{max}$ is 463.51 µm, which is 30.64% lower; furthermore, the ASO-TFLANN’s $\epsilon_{fs}$ is smallest among the four frequencies of the four methods, which is only 0.27%; taking the optimization effect of SCFLANN as a benchmark, $\epsilon_{fs}$ is reduced by 93.7%, 77.5%, and 30.6% after using ASO-TFLANN, ASO-SCFLANN, and TFLANN, in that order.

Figure 18c shows the comparison of output error images of different methods at 30 kHz. From Figure 18c, Table 4 and Table 5, it can be seen that when the primary excitation is 30 kHz, the $\epsilon_{max}$ of the four methods, ASO-TFLANN, ASO-SCFLANN, TFLANN, and SCFLANN, are 89.98 µm, 90.64 µm, 505.47 µm, 90.64 µm, 505.47 µm, 968.15 µm; $\epsilon_{fs}$ computed by ASO-TFLANN and ASO-SCFLANN are very similar to each other, respectively, 0.56% and 0.57%; the optimization effect of TFLANN is the worst, with the $\epsilon_{fs}$ of only 3.15%, which is 47.8% lower than that of SCFLANN.

Figure 18d compares output error images of different methods at 50 kHz. As can be seen from Figure 18d, Table 4 and Table 5, when the primary excitation is 50 kHz, the $\epsilon_{max}$ of ASO-TFLANN and ASO-SCFLANN are 244.41 µm, 293.65 µm, respectively, which are lower than that of the value of SCFLANN, 902.14 µm; and ASO-TFLANN’s $\epsilon_{max}$ is the smallest, and the calculated $\epsilon_{fs}$ is only 1.53%; TFLANN outputs the largest $\epsilon_{max}$, which is 1562.04 µm, and higher than SCFLANN’s 902.14 µm. Using SCFLANN’s optimization effect as a benchmark, ASO-TFLANN’s and ASO-SCFLANN’s $\epsilon_{fs}$ decrease by 72.9% and 67.4%, respectively, while TFLANN’s $\epsilon_{fs}$ increases by 73.1%. It is found that when the primary excitation is 50 kHz, the errors of all four methods are larger than 10 kHz, 20 kHz, and 30 kHz, which occurs because the position of the voltage zero point of the NC-LVDT is subsequently shifted when the frequency is increased. Taking the voltage origin position at 10 kHz as a reference, the 20 kHz offset is +0.075 mm, the 30 kHz offset is +0.1545 mm, and the 50 kHz offset is 1.775 mm. After cutting down the offsets and re-running the analog test simulation, the error range of the 50 kHz frequency is narrowed down to within ±60 µm, as shown in Figure 19.

From Figure 18a–d, in the region of small displacement (−4 mm to 4 mm), the output errors of each method are small, especially ASO-TFLANN and ASO-SCFLANN show a smoother error trend. However, when the displacement increases to near the ends (±8 mm), the error gradually increases, especially the error fluctuation of SCFLANN is larger. This indicates that as the displacement increases, the nonlinear effect of the system gradually increases, leading to an increase in the error, whereas at smaller displacements, the system has stronger linear properties, and thus the methods are better able to compensate. In addition, the maximum errors all occur at displacements of 4 mm to 8 mm. This further verifies that ASO-TFLANN and ASO-SCFLANN have good nonlinear error handling capability in different displacement ranges and outperform the conventional methods in overall error control.

### 4.3. On-Line Test Validation

To assess the practicality and effectiveness of the ASO-TFLANN nonlinear compensation method, the LVDT is connected to an oscilloscope, which is then connected to a laptop. The data received by the oscilloscope from the LVDT are transmitted to the laptop for online experimental verification. The voltage–displacement test bench adjusts the displacement of the LVDT primary coil, and the output error ϵ of the model, after the series connection with the actual displacement, is presented in Table 6. The variance and mean of the error (both calculated using the absolute values of the errors) are shown in Table 7. From Table 6, when the primary excitation is 10 kHz, the maximum output error measured in the online experiment is −55.47 µm; when the primary excitation is 20 kHz, the maximum output error measured in the online experiment is −29.67 µm; when the primary excitation is 30 kHz, the maximum output error measured in the online experiment is 57.55 µm; when the primary excitation is 50 kHz, the maximum output error measured in the online experiment is 105.27 µm. The maximum errors in the online experiment at the four frequencies are smaller than those measured in the offline experiment, which proves that the introduced method can effectively improve the nonlinearity of the LVDT. When the primary excitation frequency is 50 kHz, the error is still the largest under the four frequencies; when the primary excitation frequency is 20 kHz, the error is still the smallest under the four frequencies, which is consistent with the offline simulation experiments. From Table 7, we can see that the mean and variance of the output errors are smallest at 20 kHz. The mean error at 20 kHz is 10.42 µm, and the variance is 92.06 $µm^{2}$, making it the most stable frequency compared to the other frequencies. This suggests that the system performs most consistently at 20 kHz, with minimal error and variability.

## 5. Conclusions

In this article, the ASO-TFLANN nonlinear compensation method, which integrates the ASO with the TFLANN, is proposed, and a fitness function associated with the linearity of LVDT is established. Based on validation through offline comparative simulation tests with ASO-SCFLANN and TFLANN as well as online testing, the following conclusions are drawn:Based on the benchmark function test results, it can be concluded that the ASO algorithm outperforms other algorithms in the optimization of several test functions, showing excellent global search capability and fast convergence. Specifically, ASO achieves the best results in the optimization of the F1, F2, F3, F4, F5, and F6 functions, especially in the F1 function, where the objective function value is significantly lower than that of the other algorithms, and it rapidly decreases to a value close to $10^{-45}$ in the F2 function, which shows high accuracy and excellent convergence. In addition, in the later iterations of the F3 function, ASO successfully breaks through the local optimum, further improving the optimization effect. Overall, ASO demonstrates excellent performance in dealing with complex optimization problems, clearly outperforming other algorithms such as SO, PSO, ALO, MFO, SCA, and SS.The proposed ASO-TFLANN nonlinear compensation method, compared to ASO-SCFLANN, SCFLANN, TFLANN, and other methods, minimizes $\epsilon_{max}$ and $\epsilon_{fs}$ when operating under different frequency excitations, especially when the excitation frequency is 20 kHz, $\epsilon_{max}$ is only 42.42 µm, $\epsilon_{fs}$ is only 0.27%. Compared with the sine–cosine function, the tangent function is more effective in optimizing the linearity of LVDT.The proposed ASO-TFLANN method, compared with the traditional neural network using the gradient descent method, has stronger global and local search ability and faster convergence speed. It does not easily fall into local optimum. It requires lower requirements on the objective function, suitable for a wide range of application scenarios such as discontinuous, nonlinear, discrete problems, etc., and it has better optimization design of LVDT, which can effectively improve the LVDT. The optimization design of LVDT is more effective, and can effectively improve the linearity of LVDT.Through online tests, the ASO-TFLANN nonlinear compensation method is proven to be feasible and effective, which is of practical significance for improving the linearity of LVDT.

## Figures and Tables

**Figure 1 sensors-25-01074-f001:**
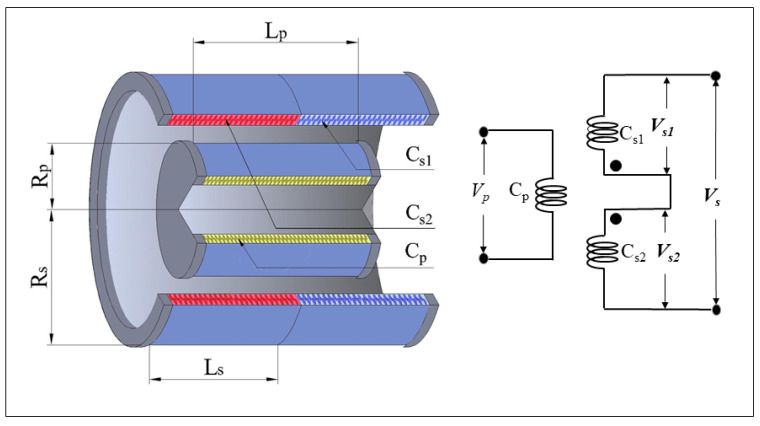
LVDT operating principle diagram.

**Figure 2 sensors-25-01074-f002:**
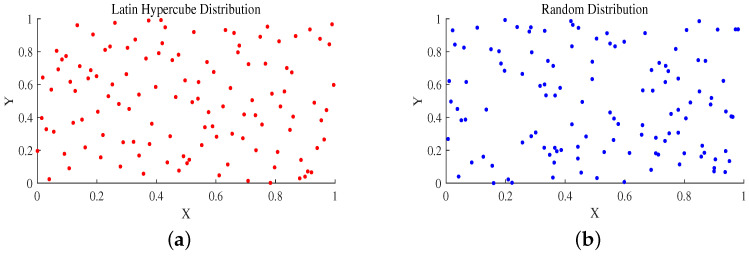
(**a**) Latin hypercube distribution; (**b**) random distribution.

**Figure 3 sensors-25-01074-f003:**
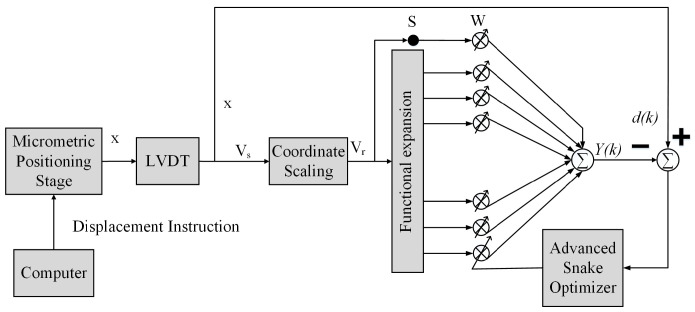
ASO-TFLANN nonlinear compensation methods.

**Figure 4 sensors-25-01074-f004:**
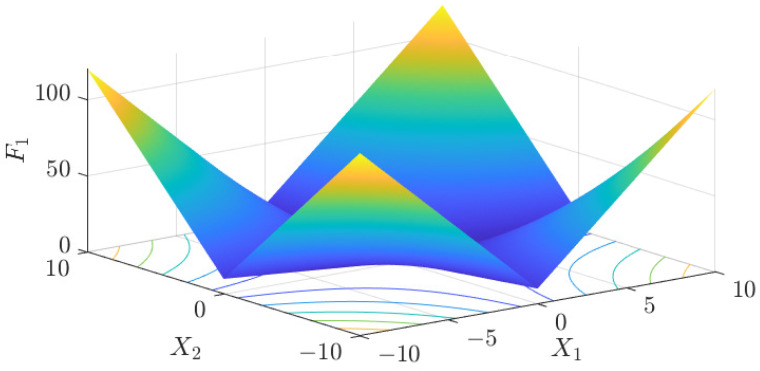
F1 three-dimensional plot of the basis function.

**Figure 5 sensors-25-01074-f005:**
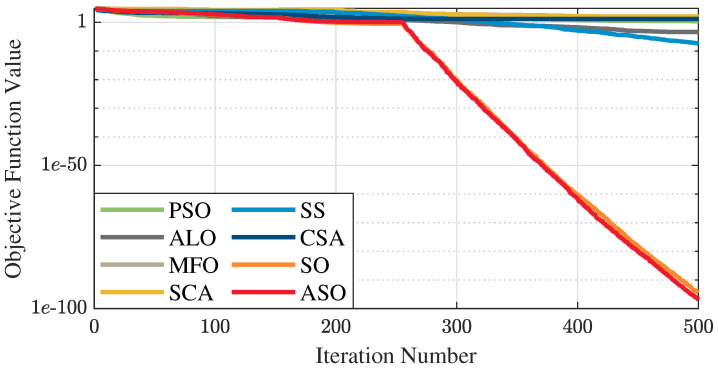
Average fitness curve of the F1 benchmark function.

**Figure 6 sensors-25-01074-f006:**
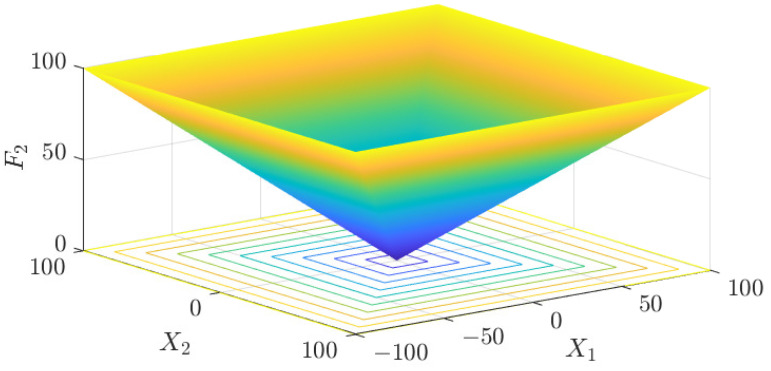
F2 three-dimensional plot of the basis function.

**Figure 7 sensors-25-01074-f007:**
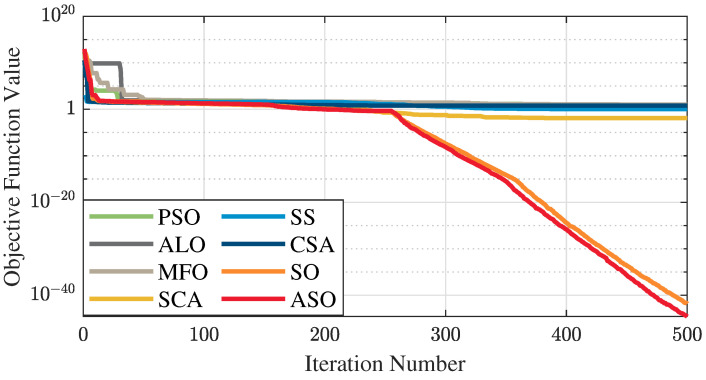
Average fitness curve of the F2 benchmark function.

**Figure 8 sensors-25-01074-f008:**
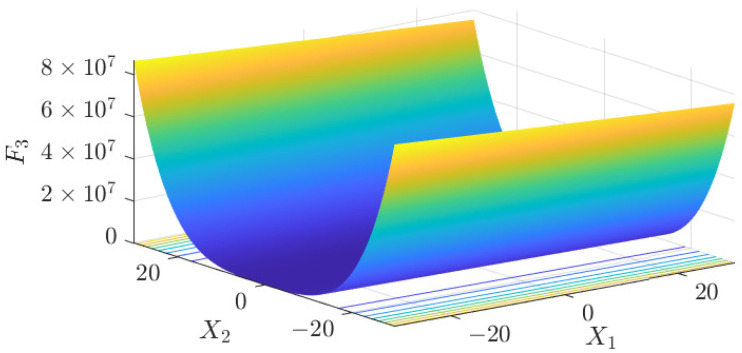
F3 three-dimensional plot of the basis function.

**Figure 9 sensors-25-01074-f009:**
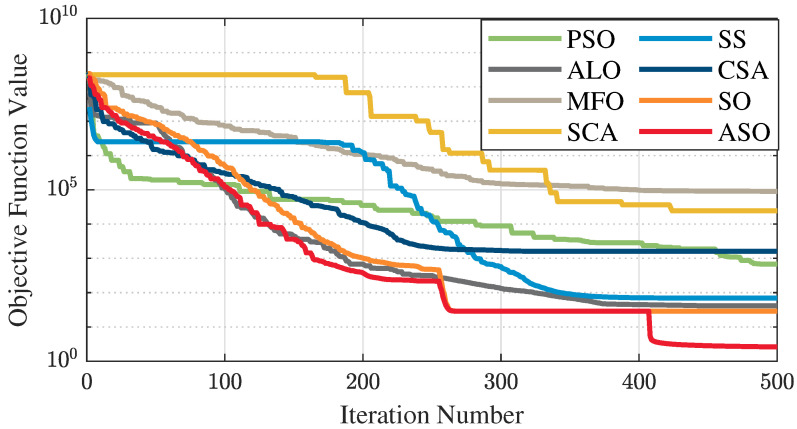
Average fitness curve of the F3 benchmark function.

**Figure 10 sensors-25-01074-f010:**
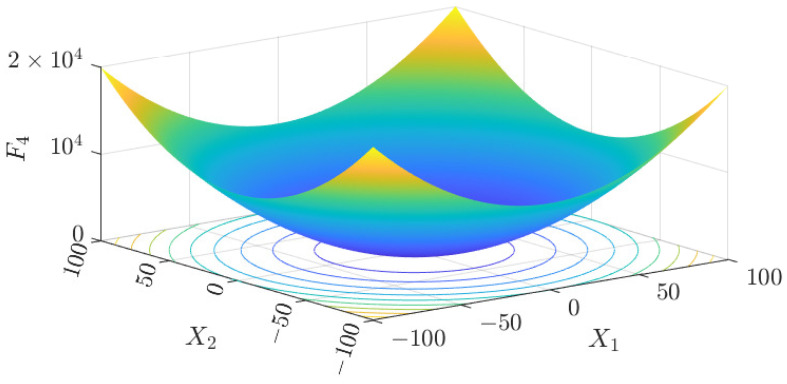
F4 three-dimensional plot of the basis function.

**Figure 11 sensors-25-01074-f011:**
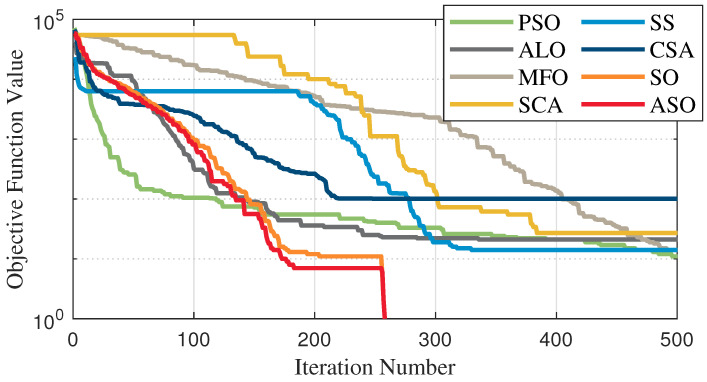
Average fitness curve of the F4 benchmark function.

**Figure 12 sensors-25-01074-f012:**
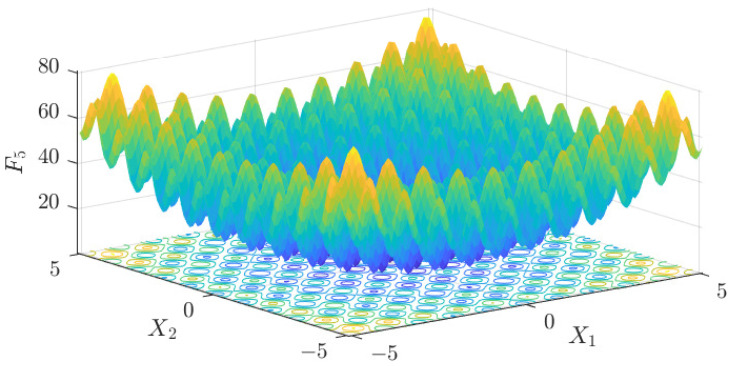
F5 three-dimensional plot of the basis function.

**Figure 13 sensors-25-01074-f013:**
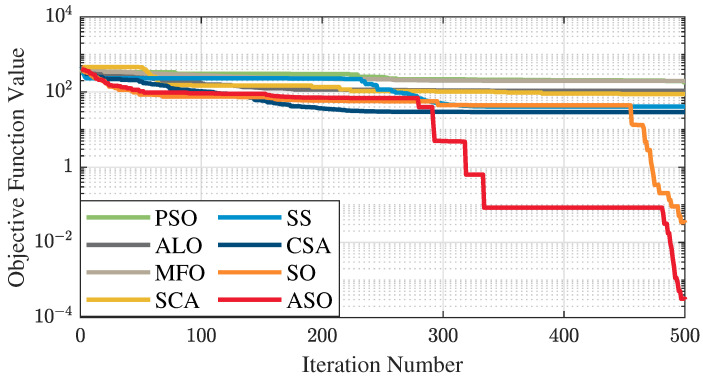
Average fitness curve of the F5 benchmark function.

**Figure 14 sensors-25-01074-f014:**
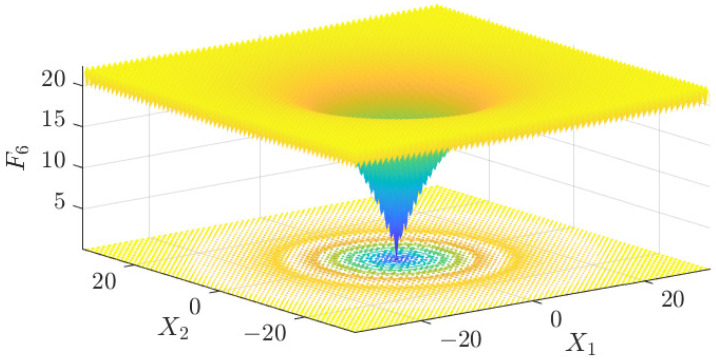
F6 three-dimensional plot of the basis function.

**Figure 15 sensors-25-01074-f015:**
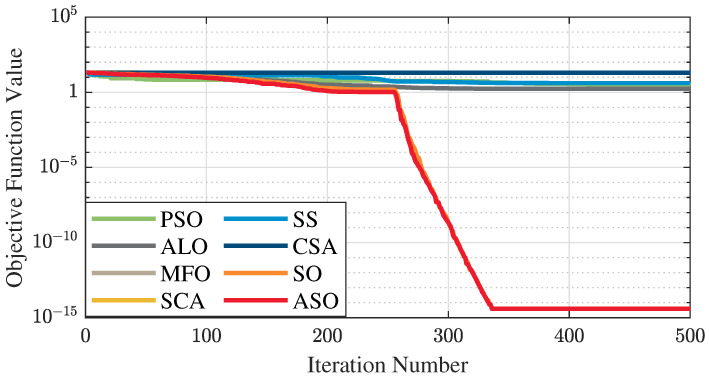
Average fitness curve of the F6 benchmark function.

**Figure 16 sensors-25-01074-f016:**
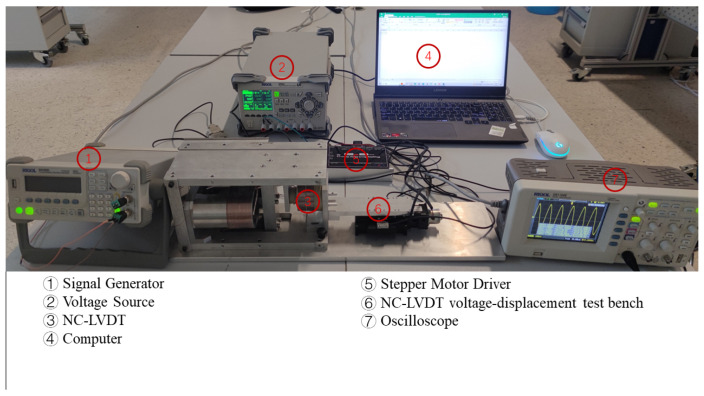
LVDT voltage–displacement test bench.

**Figure 17 sensors-25-01074-f017:**
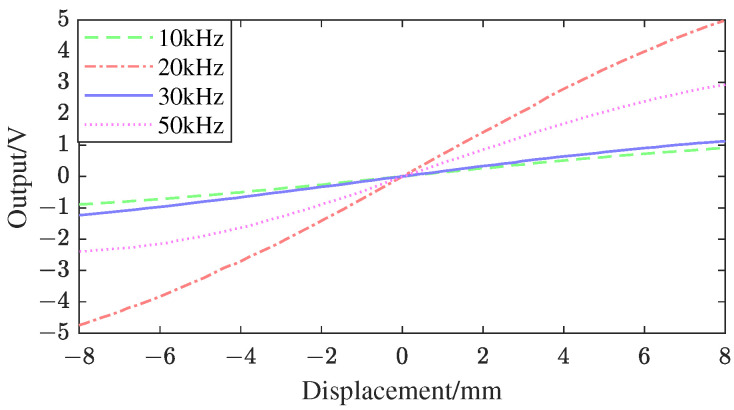
LVDT voltage−displacement test bench.

**Figure 18 sensors-25-01074-f018:**
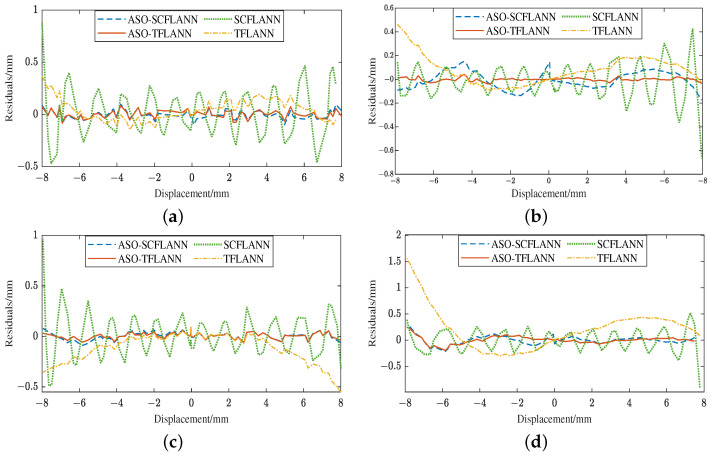
Comparison of output error images of different methods at 10 kHz, 20 kHz, 30 kHz, and 50 kHz. (**a**) Comparison of output error images of different methods at 10 kHz. (**b**) Comparison of output error images of different methods at 20 kHz. (**c**) Comparison of output error images of different methods at 30 kHz. (**d**) Comparison of output error images of different methods at 50 kHz.

**Figure 19 sensors-25-01074-f019:**
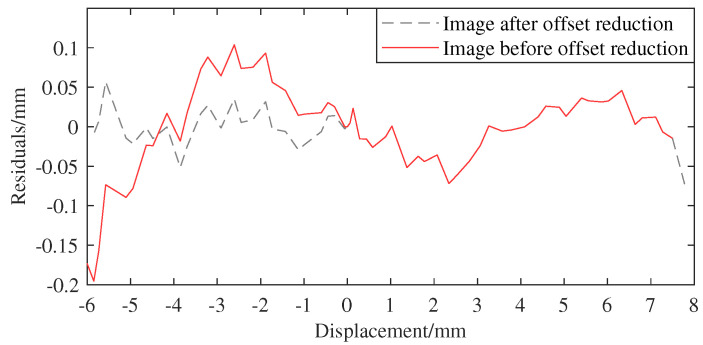
ASO−FLANN tandem LVDT output image at 50 khz before and after offset cut.

**Table 1 sensors-25-01074-t001:** Benchmark functions and their properties.

Function	Formula	Variable Scope	Theoretical-Optimum
F_1_	$\sum_{i=1}^{30}|x_{i}|+\prod_{i=1}^{30}\left|x_{i}\right|$	[−10,10]	0
F_2_	$\max|x_{i}|\;\left(1\leq i\leq 30\right)$	[−100,100]	0
F_3_	$\sum_{i=1}^{n}\left(100{\left(x_{i+1}-x_{i}^{2}\right)}^{2}+{\left(1-x_{i}\right)}^{2}\right)$	[−30,30]	0
F_4_	$\sum_{i=1}^{n}{\left(\left\lfloor x_{i}+0.5\right\rfloor \right)}^{2}$	[−100,100]	0
F_5_	$\sum_{i=1}^{n}\left(x_{i}^{2}-10\cos\left(2\pi x_{i}\right)+10\right)$	[−5.12,5.12]	0
F_6_	$\begin{array}{c} -20\exp\left(-0.2\sqrt{\frac{\sum_{i=1}^{n}x_{i}^{2}}{n}}\right)-\exp\left(\frac{\sum_{i=1}^{n}\cos\left(2\pi x_{i}\right)}{n}\right)+20+e \end{array}$	[−32,32]	0

**Table 2 sensors-25-01074-t002:** Algorithm performance—mean in the 500th generation.

Algorithm	F1	F2	F3	F4	F5	F6
PSO	2.2457	3.2882	679.4504	11	182.0177	2.6464
ALO	0.0004	9.6318	41.2399	21	108.4505	1.6982
MFO	9.9361	10.1638	90,292.8926	14	197.0225	19.9545
SCA	127.8798	0.0128	24,269.0534	27	87.9778	20.1675
SS	$4.1986\times 10^{-8}$	1.0953	69.2891	14	40.7933	4.0756
CSA	16.1634	5.9968	1596.4183	101	29.4315	19.9667
SO	$1.844\times 10^{-95}$	$7.9065\times 10^{-43}$	28.9174	0	0.0221	$3.9968\times 10^{-15}$
ASO	$4.5205\times 10^{-98}$	$1.6297\times 10^{-45}$	2.6087	0	0.0002	$3.9968\times 10^{-15}$

**Table 3 sensors-25-01074-t003:** Seventeen sets of experimental data for each frequency.

10 kHz		20 kHz		30 kHz		50 kHz	
$X_{c}$	$V_{s}$	$X_{c}$	$V_{s}$	$X_{c}$	$V_{s}$	$X_{c}$	$V_{s}$
13,297	−0.89	16,547	−4.49	27,703	−1.14	−6512	−2.89
17,632	−0.81	19,924	−4.11	24,611	−1.07	−8534	−2.77
21,538	−0.71	25,413	−3.39	20,188	−0.93	−12,301	−2.51
25,385	−0.61	30,258	−2.70	15,206	−0.78	−16,012	−2.21
29,183	−0.51	34,703	−2.00	11,770	−0.66	−20,023	−1.85
33,844	−0.36	38,559	−1.35	7060	−0.49	−24,567	−1.41
37,503	−0.26	42,676	−0.64	3386	−0.34	−28,253	−1.00
41,224	−0.13	46,209	0.00	−545	−0.18	−33,510	−0.46
45,300	0.00	50,248	0.72	−4777	0.00	−37,619	0.00
48,917	0.13	53,982	1.37	−8496	0.15	−41,545	0.46
53,050	0.26	57,898	2.04	−12,826	0.33	−46,326	0.96
57,243	0.38	62,353	2.82	−16,833	0.50	−55,549	1.77
60,944	0.50	66,066	3.40	−22,070	0.64	−59,910	1.94
65,084	0.61	70,221	3.90	−24,210	0.79	−60,549	2.09
69,466	0.73	74,595	4.57	−29,423	0.99	−64,285	2.26
74,188	0.84	76,134	4.76	−34,069	1.15	−66,087	2.30
77,298	0.91	78,052	4.97	−37,176	1.24	−68,618	2.37

**Table 4 sensors-25-01074-t004:** Four different methods of LVDT’s $\epsilon_{max}$.

Frequency	ASO-TFLANN	ASO-SCFLANN	TFLANN	SCFLANN
10 kHz	84.40	96.89	365.74	560.13
20 kHz	42.42	150.62	463.51	668.33
30 kHz	89.98	90.64	505.47	968.15
50 kHz	244.41	293.65	1562.04	902.14

**Table 5 sensors-25-01074-t005:** Four different methods of LVDT’s $\epsilon_{fs}$.

Frequency	ASO-TFLANN	ASO-SCFLANN	TFLANN	SCFLANN
10 kHz	0.61%	0.53%	2.23%	3.5%
20 kHz	0.27%	0.94%	2.89%	4.17%
30 kHz	0.56%	0.57%	3.15%	6.05%
50 kHz	1.53%	1.84%	9.76%	5.63%

**Table 6 sensors-25-01074-t006:** Output error—actual displacement table for the model after series connection. ϵ is the output error, unit is µm.

10 kHz		20 kHz		30 kHz		50 kHz	
ϵ	$X_{s}$	ϵ	$X_{s}$	ϵ	$X_{s}$	ϵ	$X_{s}$
−9.97	−7.68	15.10	−7.73	−3.28	7.50	−14.58	7.50
−18.99	−6.76	−29.67	−6.15	21.45	6.50	3.00	6.64
−55.47	−5.79	−5.37	−5.62	5.89	5.69	32.85	5.56
−16.12	−4.49	26.48	−4.57	−33.64	4.53	26.01	4.58
−54.08	−3.32	−5.94	−3.68	6.00	3.59	−5.61	3.57
−8.51	−2.55	5.57	−2.46	−6.04	2.49	−61.73	2.52
32.97	−1.48	−2.69	−1.60	12.19	1.42	−37.44	1.63
17.86	−0.63	−2.33	−0.42	0.35	0.59	−26.14	0.58
4.54	0.60	2.86	0.47	57.55	−0.47	30.55	−0.45
−22.77	1.53	13.27	1.47	−19.75	−1.47	56.31	−1.73
12.56	2.58	−7.75	2.61	42.95	−2.55	73.81	−2.45
45.18	3.61	28.72	3.64	0.88	−3.56	17.87	−3.69
35.48	4.49	−4.35	4.50	−9.75	−4.70	−23.35	−4.64
12.90	5.43	8.45	5.46	−36.76	−5.53	−89.45	−5.10
18.62	6.52	−2.18	5.77	−0.63	−6.32	−103.24	−6.48
−19.69	7.53	−5.92	6.32	13.78	−7.48	105.27	−7.43

**Table 7 sensors-25-01074-t007:** Statistical analysis of ϵ (calculated using absolute values) for different frequencies.

Parameter	10 kHz	20 kHz	30 kHz	50 kHz
Mean	24.11	10.42	16.93	44.20
Variance	253.29	92.06	297.08	1120.23

## Data Availability

Data are contained within the article.

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
