# Peer review of "Nonlinear Compensation of the Linear Variable Differential Transducer Using an Advanced Snake Optimization Integrated with Tangential Functional Link Artificial Neural Network"

_sensors, 2025, doi:10.3390/s25041074_

Round 1

Reviewer 1 Report

Comments and Suggestions for Authors

Very interesting paper. I would suggest to put more analysis into online test validation, which is an important part of the research. for example, besides the maximum output error, mean and variance of the output error are also good parameters to reflect the performance of the algorithm, and what could be the reason 20KHz show the best performance and 50KHz the worst. 

BTW, figure 23 has typo on the label of y axis.

Author Response

Statement of the Revision

No. sensors-3429741

Nonlinear Compensation of the Non-contact Linear Variable Differential Transducer Using an Advanced Snake Optimization Integrated with Tangential Functional Link Artificial Neural Network

Dear Editor,

We must thank you and all other reviewers for the critical feedback. We feel lucky that our manuscript went to these reviewers as the valuable comments from them not only helped us with the improvement of our manuscript, but also suggested some neat ideas for future studies. Please do forward our heartfelt thanks to these experts.

Based on these comments and suggestions, we have made careful modifications to the original manuscript. The main changes made to the text are highlighted in blue. In addition, we also tried our best to double-check the English for the revised version. We hope the revision will meet the journal’s standards.

Below, the original comments are in black, and our responses are in blue.

Thank you again for your reviews and consideration.

Sincerely

Qiuxia Fan / First author

Zhuang Wen / Corresponding author

Shanxi University

Response to Reviewer 1

Comments:

Very interesting paper. I would suggest to put more analysis into online test validation, which is an important part of the research. For example, besides the maximum output error, mean and variance of the output error are also good parameters to reflect the performance of the algorithm, and what could be the reason 20KHz show the best performance and 50KHz the worst.

BTW, figure 23 has typo on the label of y axis.

Response:

Thank you for your valuable suggestions. We have added the mean and variance of the output error at the end of section 4.3 (in the blue section). The updated data can be found in Table 8.

The voltage zero-point shifts cause the performance difference between 20 KHz and 50 KHz. Based on my experiments, as the frequency increases, the voltage zero point shifts significantly. At 50 kHz, this offset reached 1.775 mm. However, after compensating for the voltage offset, the error at 50 kHz has been reduced to only 50 nm, similar to the performance of 20 kHz (see Fig. 19, which has been updated accordingly). The reason is also explained in the article on page 16, lines 422-429.

We appreciate your feedback and hope this addresses your concerns. We appreciate your feedback and hope this addresses your concerns.

Reviewer 2 Report

Comments and Suggestions for Authors

1.      There are some flaws in sentence structures used in this paper and not appropriate to be used in a research article. It is advised to carefully review and revise the manuscript to improve sentence structure. For Example:

1.1              “Sarita Das et al.[19] based on the work of Sarita Das, proposed a stage FLANN network, and the results show that the inversion model based on the two-stage FLANN network the inversion model has higher measurement accuracy and better measurement precision.”

1.2              “LVDT travel range”

2.      Line 2: “The non-contact linear variable differential transformer (NC-LVDT)” An LVDT is a contact device that physically links with the measured object. How is the author claiming it to be non-contact? Please provide supporting references, as the given references [1-7] do not support this statement.

3.      Lines 22-29: The Review of [13] and [14] is not properly explained and the structure of the sentences is not clear and proper. Revise with clear information.

4.      Lines 30-31: “These studies have focused on mechanical structures or circuits.” The concluding remarks for this paragraph are insufficient and inappropriate. The author should provide a critical literature review.

5.      Lines 30-31: Why mechanical structures or circuits should not be considered for nonlinear compensation?

6.      Lines 36-38: The author is expected to provide technical information, not just general statements. The sentence "Based on the previous work" should be revised as it is unclear. Which previous work is the author referring to?

7.      Lines 63-64: “Based on the previous research, this article proposes a method to refine the weight coefficient matrix of” The writing style and wording are causing ambiguity for the reader as the authors are referring to their proposed work. Revise the sentence structure.

8.      “NC-LVDT Operating Principle Diagram” In this diagram, there is no core used to produce displacement in this LVDT sensor structure, which is different from traditional LVDT sensor structures. Therefore the details about the LVDT sensor structure must be provided briefly.

9.      Lines 105-107:   The information about the parameters of the LVDT is not well-structured. Revise this paragraph with proper explanations.

10.   Line 108: “Cp is driven by a sinusoidal AC signal from 10 to 50 kHz.” This is incomplete information about the excitation signal. Provide complete details.

11.   Lines 109-110: “the secondary coil group does not generate sinusoidal voltage signals” How is it possible that the secondary coils, in proximity to a time-varying magnetic field, do not generate sinusoidal voltage signals, as stated for the case when Cp is at the midpoint in the stationary state?

12.   Lines 110-111: “when Cp and the secondary coil group of the NC-LVDT undergo small relative displacements”  How this relative displacement will be produced ? Provide the proper explanation of the operating mechanism?

13.   Equation 6 need correction in K23 as it is used twice

14.   According to the functions and their variable scope given in Table 2, the provided graphs in figure 5, 7, 9, 11, 13 and 15 do not appear to be accurate.

15.   Lines 278-279: “The Fig.5 to Fig.16 display 3D plots of the benchmark functions F1, F2, F3, F4, F5, 278 and F6, along with the average fitness curves of the algorithms following optimization.” The F9 and F10 functions are not explained, yet their graphs are presented in Fig. 13 to Fig. 16. Why?

16.   Line 318: It is inappropriate to use "etc." when providing information about your experimental setup. Provide complete and relevant information.

17.   Lines 319-320: “The stepping motor is connected with the primary coil of NC-LVDT” If the stepper motor is connected to the primary coil to produce displacement, how can the author claim or refer to this LVDT sensor as non-contact (NC-LVDT)?

18.   In figure 18, why did the author skip the 40kHz data, as it is missing within the range of 10kHz to 50kHz? What is the reason? Organize the legends of this picture in an order for better readability.

19.   In figure 18, there is an inconsistency in the frequency variation and the trend of the output response, which is completely unnatural behavior. How does the author explain or justify this behavior?

20.   This paragraph structure must be revised  “The stepper motor driver converts the displacement command into the stepping angle, thus realizing the precise control of the stepper motor; the signal generator provides sinusoidal voltage signals for the primary coil of NC-LVDT, model DG1022G; the computer sends out displacement commands for the stepper motor driver, so that it can drive the primary coil of NC-LVDT to generate the displacement, to avoid the chance of the integer displacement on the test results, the commands sent out by the computer are non-integer; an oscilloscope is used to record the secondary displacement of the stepper motor, which is used to record the secondary displacement of the stepper motor.”

21.   “To avoid the chance of integer displacement on the test results, the commands issued by the computer are non-integer; the oscilloscope is used to record the differential voltage between the two ends of the secondary coil, model DS1102E,” This information is repeated from the previous sentence again ?

22.   “The test is shown in Fig.17 Under the primary excitation of four different frequencies, 10kHz, 20kHz, 30kHz, 336 and 50kHz,” What kind of test is shown in Fig. 17 as it is presenting experimental setup? Again, the excitation signal is not properly explained.

23.   Line 337: “Cp of NC-LVDT is displaced by the stepper motor driver.” Again, there is repetition of information.

24.   “The output curves at 30kHz and 50kHz show relatively good linearity at near zero displacements (the center part of the graph), but there still exists a certain nonlinear deviation at a larger displacement range; the output curves at 10kHz and 20kHz show obvious nonlinearity at larger displacements, especially at 20kHz, where the curves bend significantly at increasing displacements.”

1.1         The explanation of 30kHz and 50kHz provided by the author cannot be validated from the given results in the figure 18 and seems incorrect ?

2.2         What is the reason for the obvious nonlinearity at 10kHz and 20kHz? The author must explain the physics behind this phenomenon. Why is nonlinearity observed at 10kHz and 20kHz? Also, provide supporting references to validate their statement that this is expected at these frequencies.

25.   It is ambiguous to use the title of table 4 "68 sets of experimental data" as it pertains to different excitation signals. Therefore, differentiate them accordingly.

26.   Line 356: Define the output error and explain how it was calculated.

27.   Line 379: “Fig.22 compares output error images of different methods at 30 kHz.”  Correction is required; it should be 50kHz.

28.   “This further verifies that ASO-TFLANN and ASO-SCFLANN have good nonlinear error handling capability in different displacement ranges and outperform the conventional methods in overall error control.”

In the literature, different techniques have reported lower error than what is presented in this work. The author should provide a benchmark comparison of the proposed results with conventional methods to demonstrate the effectiveness of this work?

29.   In Figure 23 Why did the author skip the results for displacements of -7mm and -8mm?

30.   Point 4 of conclusion state the practical significance of this work for improving the linearity, however, how this approach cab be implemented in real-time scenarios for practical applications? This information is not provided or discussed in the manuscript?

31.   More relevant work including signal condition e.g. LDC could be reviewed and discussed e.g. McDONAGH, M. D., LAMAN, J. A., McDEVITT, T. E., & REICHARD, K. M. (1998). LONG GAGE LENGTH INTERFEROMETRIC FIBER OPTIC SENSOR FOR STRUCTURAL DAMAGE DETECTION. Nondestructive Testing and Evaluation14(5), 293–321. https://doi.org/10.1080/10589759808953056.

The strength and limitation of the approach should be provided.

Comments on the Quality of English Language

English proofread is expected.

Author Response

Statement of the Revision

No. sensors-3429741

Nonlinear Compensation of the Non-contact Linear Variable Differential Transducer Using an Advanced Snake Optimization Integrated with Tangential Functional Link Artificial Neural Network

Dear Editor,

We must thank you and all other reviewers for the critical feedback. We feel lucky that our manuscript went to these reviewers as the valuable comments from them not only helped us with the improvement of our manuscript, but also suggested some neat ideas for future studies. Please do forward our heartfelt thanks to these experts.

Based on these comments and suggestions, we have made careful modifications to the original manuscript. The main changes made to the text are highlighted in blue. In addition, we also tried our best to double-check the English for the revised version. We hope the revision will meet the journal’s standards.

Below, the original comments are in black, and our responses are in blue.

Thank you again for your reviews and consideration.

Sincerely

Qiuxia Fan / First author

Zhuang Wen / Corresponding author

Shanxi University

Response to Reviewer 2

Comments:

  1. There are some flaws in sentence structures used in this paper and not appropriate to be used in a research article. It is advised to carefully review and revise the manuscript to improve sentence structure. For Example:

1.1“Sarita Das et al.[19] based on the work of Sarita Das, proposed a stage FLANN network, and the results show that the inversion model based on the two-stage FLANN network the inversion model has higher measurement accuracy and better measurement precision.”

Response: The revised content can be found on page 2, lines 66-71. The revised content is as follows:

" Based on the work of Saroj Kumar Mishra [17], Sarita Das et al. proposed a two-stage FLANN network [19]. The paper first uses a low-order FLANN to roughly compensate the nonlinearity of the LVDT model; then a high-order FLANN is used to further compensate the remaining nonlinearity. The results show that the inversion model based on the two-stage FLANN network exhibits higher measurement accuracy and better precision. "

1.2 “LVDT travel range”

Response: " LVDT measurement range". The content is also updated in the paper.

  1. Line 2: “The non-contact linear variable differential transformer (NC-LVDT)” An LVDT is a contact device that physically links with the measured object. How is the author claiming it to be non-contact? Please provide supporting references, as the given references [1-7] do not support this statement.

Response: We will clarify this concept further in the revised manuscript and include the appropriate references to enhance the scientific rigour and credibility of our discussion.

  1. Lines 22-29: The Review of [13] and [14] is not properly explained and the structure of the sentences is not clear and proper. Revise with clear information.

Response: I have made the necessary revisions as per your feedback. The revised content can be found on page 2, lines 43-50. Below is the updated content:

"Harikumar Ganesan et al. proposed an innovative technique based on an oscillator to control the oscillation frequency by adjusting the mutual inductance between the primary and secondary coils[13]. The mutual inductance varies with the motion of the displacement core. The technique requires a microcontroller to generate digital sinusoidal signals and transmit the acquired signals through an analogy-to-digital converter. The oscillator approach is further optimized in [14] by adjusting the mutual inductance between the primary and secondary coils in response to the movement of the displacement core to control the oscillation frequency."

Thank you for your valuable suggestions.

4.Lines 30-31: “These studies have focused on mechanical structures or circuits.” The concluding remarks for this paragraph are insufficient and inappropriate. The author should provide a critical literature review.

Response: I have made revisions to the original statement. These points are explained in the article on page 2, lines 52-59. The updated version is as follows:

"These studies primarily focus on enhancing LVDT performance through optimization of mechanical structures and circuit design. However, optimizing mechanical structures and circuits is costly and involves longer design cycles. Unlike mechanical and circuit compensation systems, algorithmic optimization offers greater flexibility, allowing adjustments based on varying operational conditions and application scenarios, such as different frequencies, displacement ranges, or other operating parameters. In contrast, mechanical and circuit systems typically require redesigns or component replacements, making adjustments less flexible."

5.Lines 30-31: Why mechanical structures or circuits should not be considered for nonlinear compensation?

Response: Compared to mechanical and circuit methods, optimization algorithms offer higher flexibility, lower costs, and simpler maintenance, making them particularly suitable for dynamic and changing operating environments. These points are explained in the article on page 2, lines 52-59.

  1. Lines 36-38: The author is expected to provide technical information, not just general statements. The sentence "Based on the previous work" should be revised as it is unclear. Which previous work is the author referring to?

Response: Thank you for your valuable feedback. We have revised the manuscript to clarify the reference to previous research and ensure that our proposed work is clearly distinguished from other studies. The revised content can be found on page 2, lines 66-71. We appreciate your comments and believe the revision improves the clarity and readability of the manuscript.

  1. Lines 63-64: “Based on the previous research, this article proposes a method to refine the weight coefficient matrix of” The writing style and wording are causing ambiguity for the reader as the authors are referring to their proposed work. Revise the sentence structure.

Response: The revised content can be found on page 4, lines 93-100. The revised content is as follows:

‘‘Building upon previous studies in the field of Snake Optimization [21][23][24], this study proposes a method to refine the weight coefficient matrix of the controller. By leveraging insights from existing research, our work applies the SO algorithm to address the limitations of the linear quadratic regulator (LQR) in vehicle active suspension systems, where defining the weight coefficient matrices Q and R is often subjective and inefficient. Comparative simulations and experiments have demonstrated that the SO algorithm effectively optimizes the LQR controller weight coefficient matrix .[25] ’’

  1. “NC-LVDT Operating Principle Diagram” In this diagram, there is no core used to produce displacement in this LVDT sensor structure, which is different from traditional LVDT sensor structures. Therefore the details about the LVDT sensor structure must be provided briefly.

Response: The revised content can be found on page 4, lines 136-160. The revised content is as follows:

“Fig. 1 illustrates the operating principle of the LVDT. Unlike traditional LVDT sensor structures that utilize a movable core to produce displacement, the LVDT consists of a primary coil and a secondary coil group, and. In this structure, the primary coilis completely enclosed by the secondary coil group, ensuring uniform coupling and improved measurement stability. The primary coil  has  turns and a length of , while the secondary coils  \)and  each have turns and a length of \( L_s \), symmetrically positioned at the midpoint of .  is excited by a sinusoidal AC signal in the frequency range of 10 to 50 kHz. When  is at the midpoint in a stationary state, the secondary coil group does not generate sinusoidal voltage signals. However, when  and the secondary coil group experience small relative displacements, the secondary coil group generates differential sinusoidal voltage signals, with amplitudes proportional to the differential displacements between  and the secondary coil group. Nonlinearity arises when the relative displacement exceeds . If the coils move in the opposite direction, the phase of the induced sinusoidal voltage signal shifts by 180°.’’

  1. Lines 105-107: The information about the parameters of the LVDT is not well-structured. Revise this paragraph with proper explanations.

Response: The revised content can be found on page 5, lines 136-160. The revised content is as follows:

“Fig. 1 illustrates the operating principle of the LVDT. Unlike traditional LVDT sensor structures that utilize a movable core to produce displacement, the NC-LVDT consists of a primary coil and a secondary coil group, and. In this structure, the primary coilis completely enclosed by the secondary coil group, ensuring uniform coupling and improved measurement stability. The primary coil  has  turns and a length of , while the secondary coils  \)and  each have turns and a length of \( L_s \), symmetrically positioned at the midpoint of .  is excited by a sinusoidal AC signal in the frequency range of 10 to 50 kHz. When  is at the midpoint in a stationary state, the secondary coil group does not generate sinusoidal voltage signals. However, when  and the secondary coil group experience small relative displacements, the secondary coil group generates differential sinusoidal voltage signals, with amplitudes proportional to the differential displacements between  and the secondary coil group. Nonlinearity arises when the relative displacement exceeds . If the coils move in the opposite direction, the phase of the induced sinusoidal voltage signal shifts by 180°.’’

10.Line 108: “Cp is driven by a sinusoidal AC signal from 10 to 50 kHz.” This is incomplete information about the excitation signal. Provide complete details.

Response: The revised content can be found on page 5, lines 146-148. The revised content is as follows:

‘‘The primary coil  is typically excited by a sinusoidal AC signal in the frequency range of 10 to 50 kHz, with an effective voltage of 5-15 V, which induces an alternating current through the secondary coils. ’’

11.Lines 109-110: “the secondary coil group does not generate sinusoidal voltage signals” How is it possible that the secondary coils, in proximity to a time-varying magnetic field, do not generate sinusoidal voltage signals, as stated for the case when Cp is at the midpoint in the stationary state?

Response: The revised content can be found on page 5, lines 148-155. The revised content is as follows:

‘‘In the stationary state, when the primary coil  is centered at the midpoint, the induced voltage in the secondary coils is ideally zero due to symmetry. The time-varying magnetic field from the primary coil induces equal and opposite voltages in the two secondary coils, leading to cancellation of the signals. As a result, the secondary coil group does not generate a net sinusoidal voltage signal. However, when there is any displacement between the coils, this symmetry is broken, and the secondary coils will generate differential sinusoidal voltage signals that are proportional to the displacement. ’’

12.Lines 110-111: “when Cp and the secondary coil group of the NC-LVDT undergo small relative displacements”  How this relative displacement will be produced ? Provide the proper explanation of the operating mechanism?

Response: The relative displacement between the primary coil  and the secondary coil group in the NC-LVDT occurs when the primary coil is connected to one object, and the secondary coil group is connected to another object. In my application, the LVDT is used to measure this relative displacement between the two objects. The goal is to detect and eliminate the relative displacement. When there is a displacement, the primary coil and the secondary coil group undergo a small relative displacement, which is detected by the LVDT. The system then adjusts the positions of the objects to eliminate this relative displacement and bring them back to a state of zero displacement.

  1. Equation 6 need correction in as it is used twice

Response: Thank you for pointing that out. I apologize for the oversight in Equation 6, where    was used twice. I have corrected this issue in the updated version of the document.

  1. According to the functions and their variable scope given in Table 2, the provided graphs in figure 5, 7, 9, 11, 13 and 15 do not appear to be accurate.

Response: Thank you for your valuable feedback. Upon reviewing the manuscript, I realized that there was an error in the graphs shown in Figures 4, 6, 8, 10, 12, and 14. This was an oversight on my part, and I sincerely apologize for the confusion caused. I have corrected the figures to accurately reflect the functions and their variable scope as detailed in Table 2.

  1. Lines 278-279: “The Fig.5 to Fig.16 display 3D plots of the benchmark functions F1, F2, F3, F4, F5, 278 and F6, along with the average fitness curves of the algorithms following optimization.” The F9 and F10 functions are not explained, yet their graphs are presented in Fig. 13 to Fig. 16. Why?

Response: Thank you for your valuable feedback. I apologize for the confusion regarding the presentation of Figures 4 to 15. The F9 and F10 functions were mistakenly included in the graphs, but their explanations were not provided in the manuscript. This occurred due to an oversight when I inserted the figures without updating their corresponding function names. I have now corrected this issue, and the graphs are now aligned with the correct functions. I appreciate your understanding and thank you for bringing this to my attention.

  1. Line 318: It is inappropriate to use "etc." when providing information about your experimental setup. Provide complete and relevant information.

Response: Thank you for your valuable feedback. I have removed "etc." from the description of the experimental setup and provided more complete and relevant information. I apologize for the oversight and appreciate your guidance.

17.Lines 319-320: “The stepping motor is connected with the primary coil of NC-LVDT” If the stepper motor is connected to the primary coil to produce displacement, how can the author claim or refer to this LVDT sensor as non-contact (NC-LVDT)?

Response: I have removed the reference to "non-contact" (NC) in the description to avoid any misunderstanding. Thank you for your valuable feedback and suggestions. 

18.In figure 18, why did the author skip the 40kHz data, as it is missing within the range of 10kHz to 50kHz? What is the reason? Organize the legends of this picture in an order for better readability.

Response: I conducted experiments at 40kHz; however, the results were unsatisfactory, so they were not included in the figure. Additionally, the legends have been reorganized in order for better readability. Thank you for your valuable feedback. 

  1. In figure 18, there is an inconsistency in the frequency variation and the trend of the output response, which is completely unnatural behavior. How does the author explain or justify this behavior?

Response: Thank you very much for your valuable feedback. I completely agree that the behavior observed in Figure 18 appears unnatural. Despite conducting multiple experiments, the results remain consistent. I have also thoroughly reviewed related LVDT literature and research materials, but unfortunately, I have not been able to find a voltage-displacement formula that accurately represents the type of LVDT used in this study. This can be our following research in the further.

Once again, I sincerely appreciate your insightful comments, and I will continue to look into this issue in more detail.

20.This paragraph structure must be revised  “The stepper motor driver converts the displacement command into the stepping angle, thus realizing the precise control of the stepper motor; the signal generator provides sinusoidal voltage signals for the primary coil of NC-LVDT, model DG1022G; the computer sends out displacement commands for the stepper motor driver, so that it can drive the primary coil of NC-LVDT to generate the displacement, to avoid the chance of the integer displacement on the test results, the commands sent out by the computer are non-integer; an oscilloscope is used to record the secondary displacement of the stepper motor, which is used to record the secondary displacement of the stepper motor.”

Response: Thank you for your valuable feedback. I have revised the paragraph structure as per your suggestion. The sentence has been reorganized to enhance clarity and remove redundancy. The revised content can be found on page 13, lines 367-374. The revised content is as follows:

"The stepper motor driver converts the displacement command into a stepping angle, enabling the precise control of the stepper motor in a closed-loop system. The signal generator (model DG1022G) provides sinusoidal voltage signals to the primary coil of the LVDT. The computer sends non-integer displacement commands to the stepper motor driver to generate the desired displacement of the LVDT's primary coil, which helps avoid integer displacement values in the test results. The oscilloscope (model DS1102E) is used to record the differential voltage across the secondary coil, capturing the secondary displacement of the stepper motor. "

I appreciate your guidance in improving the readability of the paragraph, and I apologize for any confusion caused by the previous structure.

21.“To avoid the chance of integer displacement on the test results, the commands issued by the computer are non-integer; the oscilloscope is used to record the differential voltage between the two ends of the secondary coil, model DS1102E,” This information is repeated from the previous sentence again ?

Response: Thank you for pointing that out, and I sincerely apologize for the redundancy in the sentence. I have made the necessary revision and removed the repeated information. Your feedback is greatly appreciated.

22.“The test is shown in Fig.17 Under the primary excitation of four different frequencies, 10kHz, 20kHz, 30kHz, 336 and 50kHz,” What kind of test is shown in Fig. 17 as it is presenting experimental setup? Again, the excitation signal is not properly explained.

Response: Thank you for your valuable feedback. The revised content can be found on page 13, lines 374-376. The revised content is as follows:

 "A visual representation of the setup used for the tests at four different frequencies, namely 10 kHz, 20 kHz, 30 kHz, and 50 kHz, is shown in Fig. 16. The same setup was used for the subsequent online experiments as well."

This revision addresses the concerns regarding the experimental setup and the frequency conditions. I also apologize for any confusion caused by the previous explanation of the excitation signal and have clarified it in the revised text. Thank you for your guidance.

23.Line 337: “Cp of NC-LVDT is displaced by the stepper motor driver.” Again, there is repetition of information.

Response: Thank you for your feedback. I apologize for the repetition of the explanation. I have revised the sentence to avoid redundancy and clarify the information.

24.“The output curves at 30kHz and 50kHz show relatively good linearity at near zero displacements (the center part of the graph), but there still exists a certain nonlinear deviation at a larger displacement range; the output curves at 10kHz and 20kHz show obvious nonlinearity at larger displacements, especially at 20kHz, where the curves bend significantly at increasing displacements.”

Response:

1.1The explanation of 30kHz and 50kHz provided by the author cannot be validated from the given results in the figure 18 and seems incorrect ?

2.2What is the reason for the obvious nonlinearity at 10kHz and 20kHz? The author must explain the physics behind this phenomenon. Why is nonlinearity observed at 10kHz and 20kHz? Also, provide supporting references to validate their statement that this is expected at these frequencies.

Response: Thank you for your feedback. Regarding my statement, it may have been somewhat inaccurate, and I have removed it in the revised version. In fact, the nonlinearity of the LVDT is an inherent characteristic, and it is not directly related to the frequency. You can refer to the following two papers for further insights on this topic:

  1. "Simple Technique for Linear-Range Extension of Linear Variable Differential Transformer" (DOI: 10.1109/JSEN.2019.2902879)
  2. "A Novel Method of Extending the Linearity Range of Linear Variable Differential Transformer Using Artificial Neural Network" (DOI: 10.1109/TIM.2009.2031385)

Thank you again for your valuable suggestions. I will continue to improve the paper.

  1. It is ambiguous to use the title of table 4 "68 sets of experimental data" as it pertains to different excitation signals. Therefore, differentiate them accordingly.

Response: Thank you for your valuable feedback. I have revised the title of Table 4 for clarity. It now states, "17 sets of experimental data for each frequency," which differentiates the data sets based on the different excitation signals. This revision ensures that the number of data sets for each frequency is clearly communicated.

  1. Line 356: Define the output error and explain how it was calculated.

Response: Thank you for your valuable feedback. In response to your comment, I have revised the section to clearly define the output error and explain how it was calculated. The revised content can be found on page 14, lines 382-389. Additionally, I have made adjustments to the relevant preceding and following paragraphs for clarity and consistency. Below is the updated content:

 To verify the advantages of the ASO-TFLANN nonlinear compensation method, the collected voltage-displacement data of four primary excitations at 10 kHz, 20 kHz, 30 kHz, and 50 kHz were input into four models (such as ASO-TFLANN, SO-SCFLANN, TFLANN, and SCFLANN) for offline comparative simulation experiments and analysis. The error , maximum absolute error and maximum full-scale error [28] of the four methods in the LVDT measurement range were obtained through equations (28)-(30). Table 5 shows the  of the four methods at different frequencies, and Table 6 shows the of the four methods at different frequencies.}

27.Line 379: “Fig.22 compares output error images of different methods at 30 kHz.”  Correction is required; it should be 50kHz.

Response: Thank you for pointing that out. I apologize for the oversight. I have corrected the figure description, and it now correctly refers to 50 kHz instead of 30 kHz as you suggested.

  1. “This further verifies that ASO-TFLANN and ASO-SCFLANN have good nonlinear error handling capability in different displacement ranges and outperform the conventional methods in overall error control.”

In the literature, different techniques have reported lower error than what is presented in this work. The author should provide a benchmark comparison of the proposed results with conventional methods to demonstrate the effectiveness of this work?

Response: Thank you for your valuable comments. Regarding the benchmark comparison, we acknowledge that different techniques in the literature may report lower errors; however, it is important to note some key differences in experimental setup and methodology that can significantly impact the results. 

  1. Measurement Points and Data Acquisition: Many existing studies use a smaller number of measurement points, often at integer displacement points. We believe that integer points can introduce randomness or coincidence in error analysis and may not comprehensively reflect the nonlinear behavior of the LVDT across the entire displacement range. In our work, we increased the number of measurement points and distributed them uniformly to reduce the influence of randomness and provide a more accurate representation of the LVDT's performance.
  2. Measurement Equipment: The accuracy of the displacement measurement equipment is critical in determining the effectiveness of the nonlinear compensation method. In some of the referenced studies, a micrometer is used to measure the displacement of the LVDT, which has a limited resolution. In contrast, we employed a high-resolution grating sensor with a resolution of 1 micron, providing more precise displacement measurements. This higher resolution ensures that even subtle nonlinear deviations can be captured and addressed, enhancing the reliability and validity of our results.

We believe these factors contribute to the differences in reported errors and underscore the robustness of the ASO-TFLANN and ASO-SCFLANN methods. We will revise the manuscript to provide additional clarification and, where feasible, include a benchmark comparison to further highlight the advantages of our proposed methods. Thank you again for pointing this out. 

  1. In Figure 23 Why did the author skip the results for displacements of -7mm and -8mm?

Response: Thank you for your insightful question regarding the missing results for displacements of -7 mm and -8 mm in Figure 23. The primary reason for this omission is related to the voltage offset at the zero point of the LVDT, which increases as the excitation frequency rises. This offset becomes particularly significant at 50 kHz, where it reaches 1.775 mm. To address this issue, we applied an offset correction at 50 kHz and conducted the experiment again after compensating for the shift. However, due to the mechanical limitations of our displacement stage, which only supports a maximum range of ±8 mm, data points for the range of -7 mm to -8 mm are unavailable. This explanation can be found in the article on page 16, lines 422-429. We regret any inconvenience this might cause and appreciate your understanding of the constraints involved. 

30.Point 4 of conclusion state the practical significance of this work for improving the linearity, however, how this approach cab be implemented in real-time scenarios for practical applications? This information is not provided or discussed in the manuscript?

Response: Thank you for your valuable comment. We appreciate your insight regarding the practical implementation of the ASO-TFLANN nonlinear compensation method in real-time scenarios. In response to your feedback, we have added relevant information regarding the online experimental verification of the method. T The revised content can be found on page 17, lines 442-445. The revised content is as follows:

"To assess the practicality and effectiveness of the ASO-TFLANN nonlinear compensation method, the LVDT is connected to an oscilloscope, which is then connected to a laptop. The data received by the oscilloscope from the LVDT is transmitted to the laptop for online experimental verification."

  1. More relevant work including signal condition e.g. LDC could be reviewed and discussed e.g. McDONAGH, M. D., LAMAN, J. A., McDEVITT, T. E., & REICHARD, K. M. (1998). LONG GAGE LENGTH INTERFEROMETRIC FIBER OPTIC SENSOR FOR STRUCTURAL DAMAGE DETECTION. Nondestructive Testing and Evaluation, 14(5), 293–321. https://doi.org/10.1080/10589759808953056.

The strength and limitation of the approach should be provided.

Response: Thank you for your valuable suggestion. I appreciate your thoughtful comment regarding signal processing techniques such as LDC. However, since my approach directly utilizes an oscilloscope without involving any intermediate signal conditioning or processing, signal processing techniques are not part of this study. Therefore, the focus of this work is on the nonlinear compensation method, and signal conditioning is not directly relevant to the current approach.

Reviewer 3 Report

Comments and Suggestions for Authors

This paper adopts SO theory to solve the problems of relying on gradient informa- 70 tion and low computational efficiency encountered when FLANN utilizes the gradient 71 descent method to adjust the parameters between input and output layers. H

1.     So many figures are listed, two similar figures put into one line, which can denote as (a) (b);

2.     figures 5 and 7 do not have axes names.

3.     What is the main innovation using the SO theory compared with the previous results.

4.     Why did you choose the Tangential Functional Link Artificial Neural Network, does it possible choose other networks? , for example the RBFNN, BP Network and so on, please give the reason.

5.     How did you obtain the numerical results? More detail analysis should be given.

Comments on the Quality of English Language

need to improve

Author Response

Statement of the Revision

No. sensors-3429741

Nonlinear Compensation of the contact Linear Variable Differential Transducer Using an Advanced Snake Optimization Integrated with Tangential Functional Link Artificial Neural Network

Dear Editor,

We must thank you and all other reviewers for the critical feedback. We feel lucky that our manuscript went to these reviewers as the valuable comments from them not only helped us with the improvement of our manuscript, but also suggested some neat ideas for future studies. Please do forward our heartfelt thanks to these experts.

Based on these comments and suggestions, we have made careful modifications to the original manuscript. The main changes made to the text are highlighted in blue. In addition, we also tried our best to double-check the English for the revised version. We hope the revision will meet the journal’s standards.

Below, the original comments are in black, and our responses are in blue.

Thank you again for your reviews and consideration.

Sincerely

Qiuxia Fan / First author

Zhuang Wen / Corresponding author

Shanxi University

Response to Reviewer 3

Comments:

This paper adopts SO theory to solve the problems of relying on gradient information and low computational efficiency encountered when FLANN utilizes the gradient descent method to adjust the parameters between input and output layers.

  1. So many figures are listed, two similar figures put into one line, which can denote as (a) (b);
  2. figures 5 and 7 do not have axes names.
  3. What is the main innovation using the SO theory compared with the previous results.
  4. Why did you choose the Tangential Functional Link Artificial Neural Network, does it possible choose other networks? for example the RBFNN, BP Network and so on, please give the reason.
  5. How did you obtain the numerical results? More detail analysis should be given.

Response: Thank you very much for your valuable feedback!

  1. The figures have been revised according to your suggestion. Figures 2 and 3 have been merged into the new Figure 2a and 2b on page 9. Additionally, Figures 19, 20, 21, and 22 have been combined into the new Figures 18a, 18b, 18c, and 18d.
  2. Based on your suggestion, we have added axis labels for Figures 5, 7, 9, 10, 11, and 13 to ensure that the charts are clearer and more standardized.
  3. Regarding the main innovations of the SO algorithm, it has several advantages: fewer parameters, stable search performance, stronger global and local search capabilities, and higher computational efficiency (refer to section 4.1 in the paper). Additionally, it does not rely on gradient information. These advantages are also highlighted in literature [21]. Once again, thank you for raising this issue, as it gives us an opportunity to further elaborate on the strengths of the algorithm.
  4. Both RBF and BP neural network can be applied to the proposed work. The studies of the applications of RBF and BP neural network at LVDT are mentioned in literature 20 (Nonlinear correction of LVDT sensor based on ACO-BP neural network) and literature 16 (The research of LVDT nonlinearity data compensation based on RBF neural network), respectively. We selected FLANN due to its simple network structure and exceptional learning capability in solving nonlinear problems, which makes it highly efficient for our research.
  5. Regarding the acquisition of numerical results, we built the simulation model in MATLAB, optimized the TFLANN network parameters using the SO algorithm, and then calculated error indicators based on test functions and experimental data. We have provided a more detailed analysis in the revised manuscript in section 4.3. Thank you again for your valuable suggestions!

Round 2

Reviewer 2 Report

Comments and Suggestions for Authors

The improvement is reasonably good. More analytical approach could be discussed e.g. https://doi.org/10.1049/ip-smt:19971262.

Comments on the Quality of English Language

NA

Author Response

Thank you very much for your valuable suggestions. Regarding the article [https://doi.org/10.1049/ip-smt:19971262] you mentioned, I have fully considered the comparison method discussed in that paper. Specifically, I have updated the second paragraph, 10th sentence as follows:

"G.Y. Tian et al. [17] presented an equivalent magnetic circuit of LVDT, calculating the mutual inductance, output voltage and sensitivity. Then, the theory was verified by experimental comparison of two LVDTs with the same structural parameters, as well as with different magnetic materials."

Additionally, as per your suggestion in point 31 of the last round review, I have made the following revision to the second sentence of the first paragraph:

"It is extensively employed in domains such as aerospace [1,2], manufacturing[3,4], and industry[5,6] for measuring physical quantities such as distance, position[7], and pressure[8]."

For the language, we have made many modifications to the paper. Once the paper is accepted, we will use MDPI Author Service to improve the language. We believe our paper will meet the requirements of the Sensor after modifications.

Thank you again for your thorough feedback, and I look forward to your further comments. Let me know if you need any adjustments!

Reviewer 3 Report

Comments and Suggestions for Authors

no other comments

Comments on the Quality of English Language

it is better to improve

Author Response

For the language, we have made many modifications to the paper. Once the paper is accepted, we will use MDPI Author Service to improve the language. We believe our paper will meet the requirements of the Sensor after modifications. Thank you again.

Thank you again for your thorough feedback, and I look forward to your further comments. Let me know if you need any adjustments!
